# A lung-on-chip model of early *Mycobacterium tuberculosis* infection reveals an essential role for alveolar epithelial cells in controlling bacterial growth

Vivek V Thacker[1]*, Neeraj Dhar[1], Kunal Sharma[1], Riccardo Barrile[2†], Katia Karalis[2], John D McKinney[1]*

[1]School of Life Sciences, Swiss Federal Institute of Technology Lausanne (EPFL), Lausanne, Switzerland; [2]Emulate Inc, Boston, United States

**Abstract** We establish a murine lung-on-chip infection model and use time-lapse imaging to reveal the dynamics of host-*Mycobacterium tuberculosis* interactions at an air-liquid interface with a spatiotemporal resolution unattainable in animal models and to probe the direct role of pulmonary surfactant in early infection. Surfactant deficiency results in rapid and uncontrolled bacterial growth in both macrophages and alveolar epithelial cells. In contrast, under normal surfactant levels, a significant fraction of intracellular bacteria are non-growing. The surfactant-deficient phenotype is rescued by exogenous addition of surfactant replacement formulations, which have no effect on bacterial viability in the absence of host cells. Surfactant partially removes virulence-associated lipids and proteins from the bacterial cell surface. Consistent with this mechanism, the attenuation of bacteria lacking the ESX-1 secretion system is independent of surfactant levels. These findings may partly explain why smokers and elderly persons with compromised surfactant function are at increased risk of developing active tuberculosis.

*For correspondence:
vivekvthacker@gmail.com (VVT);
john.mckinney@epfl.ch (JDMK)

Present address: †University of Cincinnati, Cincinnati, United States

## Introduction

Early tuberculosis (TB), a respiratory infection caused by *Mycobacterium tuberculosis* (Mtb) is strongly influenced by host physiology; due to the small diameter of respiratory bronchioles only the smallest aerosol droplets containing one to two bacilli are successfully transported to the alveolar space (*Ma and Darquenne, 2011*), and the 'first contact' with a naive host is by default a single-cell interaction between an Mtb bacillus and a host cell. There is some evidence that pulmonary surfactant plays host-protective role in these early interactions (*Torrelles and Schlesinger, 2017*), but a complete understanding of the role of surfactant is difficult to obtain from animal infection models owing to the lethality of surfactant deficiency. In addition, experiments in animal models (*Collins and Orme, 1994*) cannot provide information about the dynamics of host-Mtb interactions at this early stage with sufficient spatiotemporal resolution (*Westphalen et al., 2014*; *Looney et al., 2011*). A commonly used in vitro model, infection of macrophages with Mtb (*Lerner et al., 2017*), has been used to probe the role of certain surfactant components (*Arcos et al., 2011*), but these studies cannot address the role of native surfactant secreted by alveolar epithelial cells (ATs) at an air-liquid interface (ALI), a condition that has been reported to alter Mtb physiology (*Ojha et al., 2008*).

Organ-on-chip systems recreate tissue-level complexity in a modular fashion, allowing the number of cellular components, their identity, and environmental complexity to be tailored to mimic key

**eLife digest** Tuberculosis is a contagious respiratory disease caused by the bacterium *Mycobacterium tuberculosis*. Droplets in the air carry these bacteria deep into the lungs, where they cling onto and infect lung cells. Only small droplets, holding one or two bacteria, can reach the right cells, which means that just a couple of bacterial cells can trigger an infection. But people respond differently to the bacteria: some develop active and fatal forms of tuberculosis, while many show no signs of infection. With no effective tuberculosis vaccine for adults, understanding why individuals respond differently to *Mycobacterium tuberculosis* may help develop treatments.

Different responses to *Mycobacterium tuberculosis* may stem from the earliest stages of infection, but these stages are difficult to study. For one thing, tracking the movements of the few bacterial cells that initiate infection is tricky. For another, studying the molecules, called 'surfactants', that the lungs produce to protect themselves from tuberculosis can prove difficult because these molecules are necessary for the lungs to inflate and deflate normally. Normally, the role of a molecule can be studied by genetically modifying an animal so it does not produce the molecule in question, which provides information as to its potential roles. Unfortunately, due to the role of surfactants in normal breathing, animals lacking them die. Therefore, to reveal the role of some of surfactants in tuberculosis, Thacker et al. used 'lung-on-chip' technology. The 'chip' (a transparent device made of a polymer compatible with biological tissues) is coated with layers of cells and has channels to simulate air and blood flow.

To see what effects surfactants have on *M. tuberculosis* bacteria, Thacker et al. altered the levels of surfactants produced by the cells on the lung-on-chip device. Two types of mouse cells were grown on the chip: lung cells and immune cells. When cells lacked surfactants, bacteria grew rapidly on both lung and immune cells, but when surfactants were present bacteria grew much slower on both cell types, or did not grow at all. Further probing showed that the surfactants pulled out proteins and fats on the surface of *M. tuberculosis* that help the bacteria to infect their host, highlighting the protective role of surfactants in tuberculosis.

These findings lay the foundations for a system to study respiratory infections without using animals. This will allow scientists to study the early stages of *Mycobacterium tuberculosis* infection, which is crucial for finding ways to manage tuberculosis.

aspects of the relevant physiology, such as an ALI in a lung-on-chip (LoC) (*Huh et al., 2010*). These systems have emerged as crucial tools for the replacement of animal models in drug development, toxicity testing, and personalized medicine (*Ghaemmaghami et al., 2012*; *Ronaldson-Bouchard and Vunjak-Novakovic, 2018*). A far less-explored line of enquiry has been to use them as models to study the dynamics of host-pathogen interactions in a realistic physiological setting (*Grassart et al., 2019*), where they can combine key advantages of both simpler in vitro models and animal models (*Torrelles and Schlesinger, 2017*). Here, we develop an LoC model of early TB infection and use time-lapse microscopy to study the infection dynamics for ATs and macrophages as independent sites of first contact, and the impact of surfactant on infection of ATs and macrophages under ALI conditions that mimic the alveolar environment in vivo.

## Results

### LoC model of early Mtb infection

Freshly isolated mouse ATs comprise a mix of type I cells (*Figure 1A*) and type II cells that produce normal surfactant (NS) levels (*Figure 1B*). Prolonged in vitro passage causes ATs to adopt a phenotype with deficient surfactant (DS) levels (*Figure 1C*). DS cells had reduced expression of type II markers such as *Abca3* (required for surfactant export [*Beers and Mulugeta, 2017*; *Besnard et al., 2010*]) and all four surfactant proteins *Sftpa*, *Sftpb*, *Sftpc*, and *Sftpd* (*Figure 1—figure supplement 1A*) as measured by RT-PCR. These cells also showed reduced expression of some type I markers such as the membrane proteins Aquaporin (*Aqp5*) and Podoplanin (*Pdpn*) but had elevated expression of other type I markers such as Caveolin-1 (*Cav1*) and Insulin Growth Factor Binding Protein 2 (*Igfbp2).* These observations are consistent with a majority of the cells in this populations having a

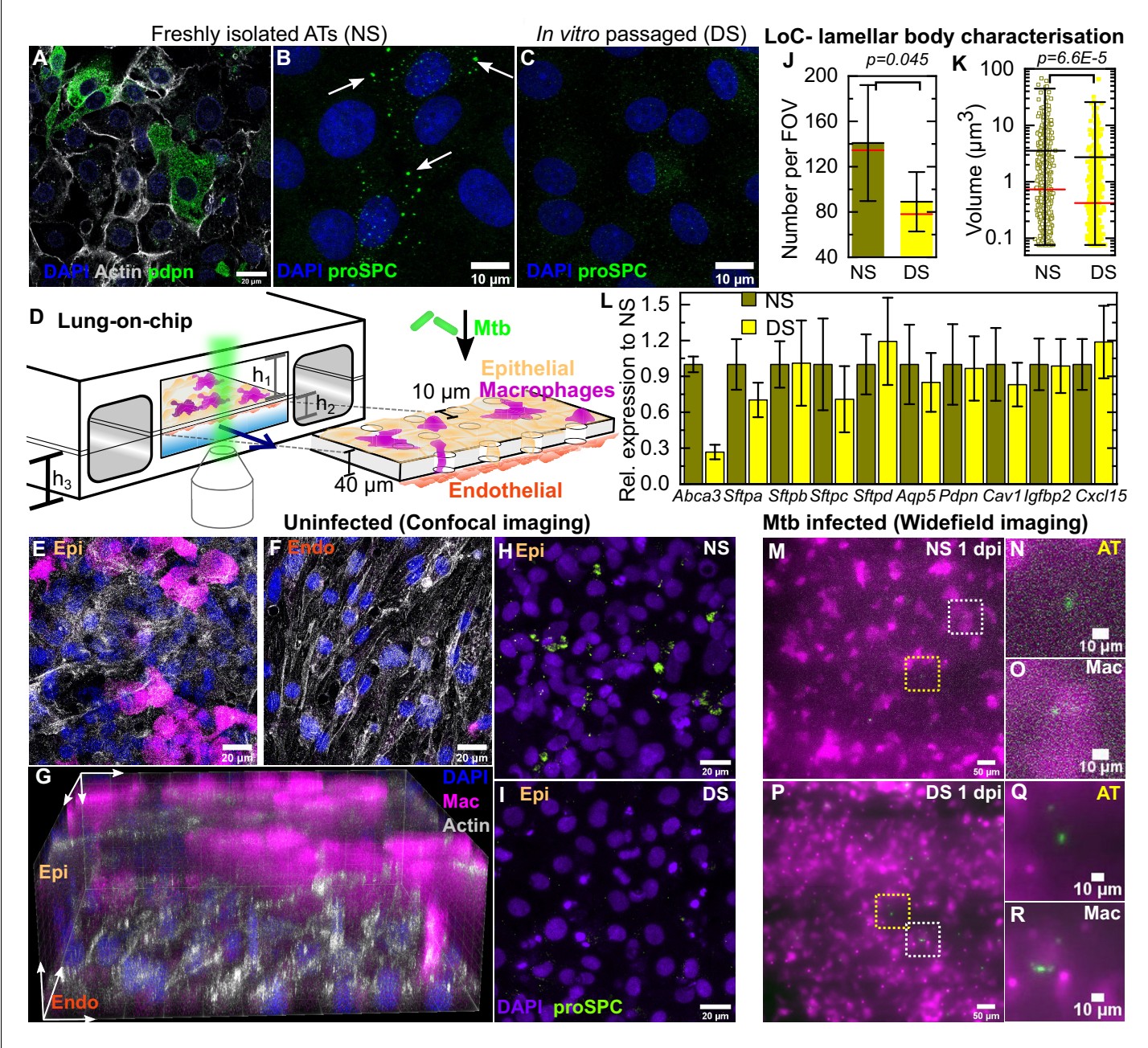

**Figure 1.** Direct observation of the role of pulmonary surfactant in a lung-on-chip (LoC) model for tuberculosis. Freshly isolated murine alveolar epithelial cells (ATs) are a mixture of (**A**) type I Pdpn-immunostained cells and (**B**) type II pro-SPC-immunostained cells containing lamellar bodies (white arrows). (**C**) In vitro passaged ATs have fewer and smaller lamellar bodies. (**D**) Schematic of the LoC model of early tuberculosis. Confluent layers of ATs and endothelial cells populate the top and bottom faces of the porous membrane that separates the air-filled 'alveolar' (upper) and liquid-filled 'vascular' (lower) compartments, creating an air-liquid interface. GFP-expressing macrophages (magenta) are added to the alveolar compartment to mimic the natural route of infection. $h_1$ = 1 mm, $h_2$ = 250 μm, $h_3$ = 800 μm. (**E-F**) Confocal microscope images of an uninfected LoC stained to visualize nuclei (blue), actin (gray), and surfactant (green, anti-pro-SPC antibody) verifies that confluency of epithelial (**E**) and endothelial (**F**) layers is maintained at the air-liquid interface over 7 days. 3D imaging reveals occupancy of some pores by macrophages (**G**). Maximum intensity projection of a Z stack from representative field of view on an LoC reconstituted with normal surfactant (NS) (**H**) or deficient surfactant (DS) (**I**) ATs shows that these phenotypes are maintained at the air-liquid interface. Data from independent 155 × 155 μm² fields of view from NS and DS LoCs (n=5 for NS and n=6 for DS) show that DS LoCs have fewer (**J**) and smaller (**K**) lamellar bodies. Mean and median values are represented by black and red bars, respectively. Whiskers represent the standard deviation in (**J**) and the 1–99 percentile interval in (**K**), (p = 0.045 and 6.6E-5, respectively). DS LoCs also have reduced expression of type II AT markers relative to NS LoCs (**L**), error bars represent the standard deviation for two technical repeats. (**M-R**) Widefield microscope images of LoCs reconstituted with NS (**M-O**) or DS (**P-R**) ATs and infected with a low dose of Mtb expressing td-Tomato (green). Images

*Figure 1 continued*

taken at 1 day post-infection (dpi) show that ATs (yellow boxes (**M, P**) and zooms (**N, Q**)) as well as macrophages (white boxes (**M, P**) and zooms (**O, R**)) can be sites of first contact. p-Values were calculated using the Kruskal-Wallis one-way ANOVA test.

The online version of this article includes the following figure supplement(s) for figure 1:

**Figure supplement 1.** qRT-PCR characterization of expression of AT markers in freshly isolated and in vitro passaged murine alveolar epithelial cells relative to *Gapdh*.

**Figure supplement 2.** Additional characterization of AT phentype on-chip.

**Figure supplement 3.** Mtb infection of ATs in vivo in the mouse model at 8 dpi.

terminal type I phenotype (*Wang et al., 2018*). DS ATs also have fewer and smaller lamellar bodies (*Figure 1—figure supplement 1B,C*).

We reconstituted LoC (schematic in *Figure 1D*) with confluent monolayers of NS or DS ATs (*Figure 1E*) and endothelial cells (*Figure 1F*) on opposite faces of a porous membrane (*Figure 1G*) and an ALI mimicking the alveolar environment. Macrophages added to the epithelial face may remain there or transmigrate across the membrane to the endothelial face (*Figure 1E–G*). Typical numbers of ATs, endothelial cells, and macrophages are given in *Table 1*, the low macrophage: AT ratio is consistent with alveolar physiology (*Weibel, 2015*). The macrophages are obtained from a mouse line that constitutively expresses GFP (false-colored magenta in all Figures unless otherwise specified) to enable unambiguous identification of these cells during live-cell microscopy. A maximum intensity projection of a field of view on the epithelial face of an LoC reconstituted with NS ATs (*Figure 1H*, additional examples in *Figure 1—figure supplement 2A–C*) and maintained for 24 hr at the ALI shows more intense pro-SPC signal than a corresponding maximum intensity projection for an LoC reconstituted with DS ATs (*Figure 1I*, additional examples in *Figure 1—figure supplement 2D–F*). LoCs reconstituted with DS ATs retain deficient surfactant expression on-chip at the ALI with fewer (*Figure 1J*) and smaller (*Figure 1K*) lamellar bodies detected across multiple fields of view. This phenotype is also reflected in the reduced expression of some type II markers, notably *Abca3,* in DS LoCs as compared to NS LoCs (*Figure 1L*, *Figure 1—figure supplement 2G*). In contrast, expression of type I markers including *Aqp5* and *Pdpn* as well as the lung chemokine *Cxcl15* was not significantly different between NS and DS LoCs at the ALI (*Figure 1L*, *Figure 1—figure supplement 2H*). This established LoCs reconstituted with DS ATs as a tool for the direct study of the role of AT-secreted pulmonary surfactant in early infection without significantly altering other aspects of AT biology. DS LoCs retained surfactant deficiency for up to 6 days at the ALI (*Figure 1—figure supplement 2I,J* vs. *Figure 1—figure supplement 2K,L*).

Inoculation of the LoC with between 200 and 800 Mtb bacilli led to infection of both macrophages (white boxes in *Figure 1M,P*, zooms in *Figure 1O,R*) and ATs (yellow boxes in *Figure 1M,P*, zooms in *Figure 1N,Q*) under both NS (*Figure 1M–O*) and DS (*Figure 1P–R*) conditions. The current paradigm in TB focuses on macrophages as sites of first infection; we therefore examined all the Mtb-infected cells isolated from the lungs of a mouse at 8 days post-infection (dpi) in an unbiased manner to ascertain if ATs also served as a site of first contact. This revealed that 7.3% of infected cells (n = 163) were CD45- pro-SPC+ type II ATs (*Figure 1—figure supplement 3A–F*, *Video 1*). We did not find instances of type I AT infection, but this likely reflects the challenges in isolating this cell type. Thus, the LoC model faithfully reproduces AT infection that also occurs in vivo (*Figure 1—*

**Table 1.** Characterization of AT, endothelial cell, and macrophage densities in deficient surfactant (DS) and normal surfactant (NS) LoCs.

| Surfactant | Cell type | Cell density$\times 10^4$/ mm$^2$ |
| --- | --- | --- |
| DS | AT | 23.5±3.9 |
| NS | AT | 26.4±6 |
| DS | Macrophage | 3.3±1.5 |
| NS | Macrophage | 3.2±1.5 |
| n/a | Endothelial | 13.6±3.7 |

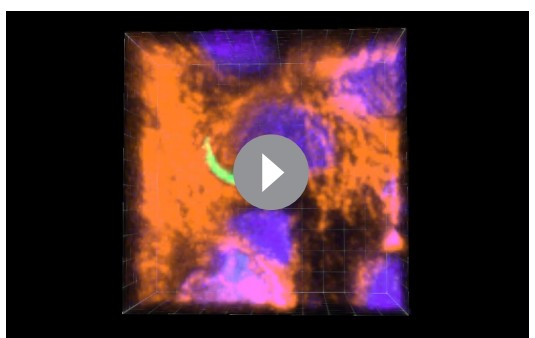

**Video 1.** 3D rotation along the Y-axis of a Z-stack highlighting the intracellular localization of Mtb within a type II AT. The field of view is 81.98 x 81.98 x 15.12 μm³. Colors: pro-SPC is labeled with the Amber LUT, Mtb is labeled with the Spring Green LUT, and CD45 is labeled with the Azure LUT.

https://elifesciences.org/articles/59961#video1

*figure supplement 3G*) albeit at a higher frequency (*Figure 1—figure supplement 3H*). This in turn enables us to study the infection dynamics in ATs and macrophages simultaneously.

## Surfactant deficiency leads to uncontrolled intracellular growth of Mtb

We used time-lapse microscopy to quantify the intracellular growth of Mtb in the host cells of first contact by measuring the total fluorescence intensity of bacteria within single infected cells over time (*Figure 2*). The refractive index differences at the ALI significantly degrades axial resolution and signal-to-noise ratios; nonetheless, we are able to identify and track individual infected cells over time. Between 3 and 5 dpi under NS (*Figure 2, A-D*, *Figure 2—video 1*) or DS (*Figure 2, E-H*, *Figure 2—video 2*) conditions, intracellular growth of Mtb is highly variable in both ATs (*Figure 2I,K*) and macrophages (*Figure 2J,L*). We used high-resolution confocal imaging of the epithelial face of an infected LoC fixed at 4 dpi to obtain direct verification that growth in both ATs and macrophages is intracellular (*Figure 2—figure supplement 1* and *Videos 2* and *3*). Plots of the logarithm of total bacterial fluorescence intensity within individual infected cells (representing the spread in growth rates) indicate that bacterial growth is exponential in ATs (*Figure 2I,K*) and macrophages (*Figure 2J,L*). However, under NS conditions, we identified substantial fraction of bacteria that show very slow growth (doubling time >168 hr) or even a decrease in fluorescence intensity over time. In the absence of a reliable live/dead marker for Mtb, we identify this as a 'non-growing fraction' (NGF). Intracellular bacterial growth is slower in macrophages compared to ATs under NS conditions but growth rates in both cell types are equivalent under DS conditions (*Figure 2M*). Compared to bacterial growth in axenic 7H9 cultures, growth in both cell types is slower under NS conditions (*Figure 2—figure supplement 2A*) but significantly *faster* under DS conditions (*Figure 2—figure supplement 2B*). We also found that Mtb grows very poorly when cultured axenically in the ALI medium (*Figure 2—figure supplement 3* vs *Figure 2—figure supplement 4B*), which provides indirect evidence that Mtb growth on-chip is likely to be intracellular. Interestingly, there is a much larger spread in growth rates and a small fraction of bacteria continue to grow rapidly even under NS conditions. We observed no spatial pattern of intracellular Mtb growth rates within the LoC in both NS and DS conditions, confirming that the observed distributions of growth rates are not due to spatial heterogeneity within the device (*Figure 2—figure supplement 5A,B*). Taken together, these results suggest that surfactant deficiency shifts the host-pathogen equilibrium in favor of Mtb, resulting in uncontrolled bacterial growth even in macrophages.

## Mtb growth rates are in good agreement with data from the mouse model obtained with a replication clock plasmid

We compared the growth rate measurements from the LoC with those obtained from *Gill et al., 2009* which uses of a plasmid-loss assay in the mouse model of infection and represents the state-of-the-art in quantitative measurements of Mtb growth in vivo. *Table 1* (*Gill et al., 2009*) in lists a mean growth rate $r$ = 0.78 and a mean death rate $\delta$ = 0.41 for Mtb replication for the period of days 1–14 post-infection in the mouse model of infection. Thus, the *net* growth rate equals $r-\delta$=0.37, which corresponds to a doubling time (or generation time) of $t_d = \frac{\ln(2)}{r-\delta} \cdot 24$ hr = 45 hr. In the notation of the current manuscript, this converts to a growth rate (h⁻¹) of 0.022, which is in good agreement with the mean or median growth rate that we report for macrophage infections with wild-type Mtb in NS conditions (*Table 2*). Notably, the LoC model provides the entire population distribution of growth rates with an approximately 200-fold higher temporal resolution.

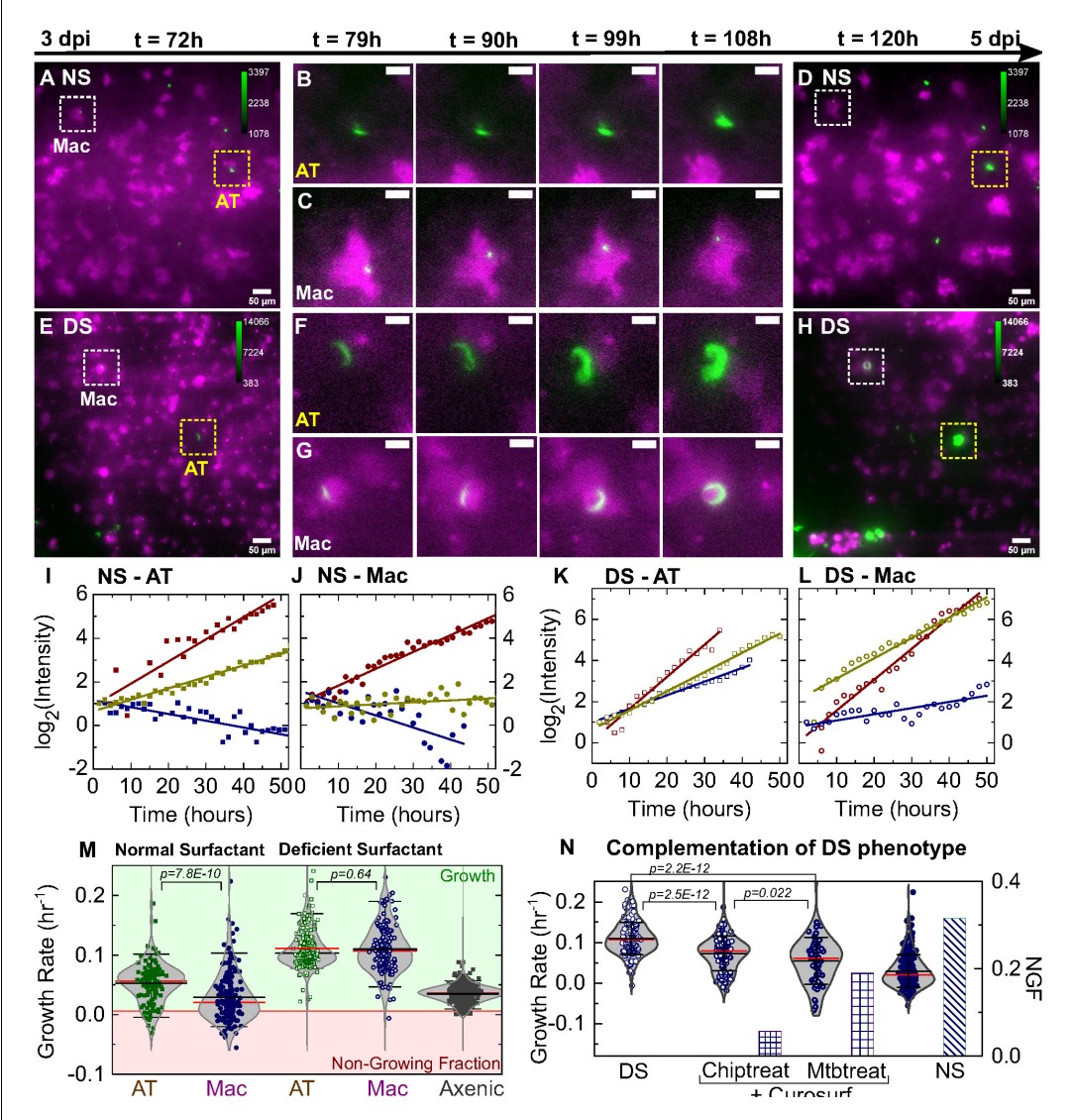

**Figure 2.** Surfactant deficiency results in uncontrolled intracellular growth of Mtb. Snapshots from live-cell imaging at 1.5–2.0 hr intervals between 3 and 5 days post-infection (dpi) in normal surfactant (NS) (**A-D**) and deficient surfactant (DS) (**E-H**) lung-on-chip (LoCs). Macrophages are false-colored magenta; Mtb is false-colored green. The calibration bar (inset in **A, D, E, H**) indicates the absolute intensities in the Mtb channel, the scales were chosen to achieve a similar saturation level in the images across surfactant conditions. Representative examples of infected AT (yellow boxes) and macrophage (white boxes) are highlighted, and zooms (**B, C, F, G**) reveal growth in both cell types over this period. Scale bar, 10 μm. (**I, J**) Plots of the logarithm of total fluorescence intensity over time confirm exponential Mtb growth for representative infections in ATs (**I, K**) and macrophages (**J, L**) under NS and DS conditions, respectively. In each case, an intracellular microcolony with growth rate close to the population maximum (red), median (yellow), and minimum (blue) is shown. The growth rate is the gradient of the linear fit. (**M**) Scatter plots of Mtb growth rates in ATs (n=122 for NS, n=219 for DS) and macrophages (n=185 for NS, n=122 for DS). Growth is significantly slower in macrophages than ATs in NS conditions (p = 7.8E-8) but not in DS conditions (p = 0.64), and is more heterogenous in both conditions compared to single-cell Mtb growth rate data from axenic microfluidic cultures. The green- and red-shaded regions indicate the growing bacteria and the non-growing fraction (NGF), respectively. (**N**) Uncontrolled growth in DS conditions can be rescued by exogenous administration of Curosurf. Scatter plots represent Mtb growth rates in macrophages in a DS LoC treated with Curosurf ('Chiptreat') or infected with Mtb preincubated with Curosurf ('Mtbtreat'). Data from DS and NS LoC infections (no Curosurf) are included for comparison. Growth attenuation for both treatments is significant relative to DS conditions as reflected by the average growth rate and the size of the NGF (n = 122 for DS and n = 121 for Chiptreat; p = 2.5E-12 and n = 63 for Mtbtreat; p = 2.2E-12 and n = 122 for DS). p-Values were calculated using the Kruskal-Wallis one-way ANOVA test.

The online version of this article includes the following video and figure supplement(s) for figure 2:

**Figure supplement 1.** Characterization of intracellular Mtb growth in a lung-on-chip (LoC) treated with surfactant (chiptreat) at 4 days post-infection.
**Figure supplement 2.** Comparison of Mtb growth rates on-chip vs. single-cell growth rates in axenic conditions.
**Figure supplement 3.** Characterization of the growth characteristics of WT and the Δicl1Δicl2 Mtb strain in ALI media.

*Figure 2 continued on next page*

*Figure 2 continued*

**Figure supplement 4.** Curosurf does not affect Mtb viability or growth in vitro.

**Figure supplement 5.** Heterogenous host-Mtb interactions are not restricted to a particular spatial niche on-chip.

**Figure 2—video 1.** Live-cell imaging over 3-5 days post-infection at the ALI for an LoC infected with WT Mtb in normal surfactant conditions and corresponding to snapshots in *Figure 2A-D*.

https://elifesciences.org/articles/59961#fig2video1

**Figure 2—video 2.** Live-cell imaging over 3-5 days post-infection at the ALI for an LoC infected with WT Mtb in deficient surfactant conditions and corresponding to snapshots in *Figure 2E-H*.

https://elifesciences.org/articles/59961#fig2video2

## Exogenous addition of surfactant restores control of Mtb growth

Although reduced surfactant secretion in DS LoCs correlates with increased Mtb replication, this shift could reflect other physiological changes that occur during in vitro passage of ATs. We therefore asked whether uncontrolled intracellular replication of Mtb in LoCs reconstituted with DS ATs could be rescued by exogenous addition of surfactant. A 1% solution of Curosurf, a pulmonary surfactant formulation comprising dipalmitoylphospatidylcholine (DPPC) and the hydrophobic surfactant proteins SP-B and SP-C, was used to treat either the Mtb or the DS LoC prior to infection. Both procedures attenuated intracellular Mtb growth and generated a non-growing fraction similar in magnitude to infected NS LoCs (*Figure 2N*, *Table 2*). Curosurf treatment affects neither Mtb viability (*Figure 2—figure supplement 4A*) nor replication in vitro in the absence of host cells (*Figure 2—figure supplement 4B*) suggesting that surfactant protects by altering the interaction of Mtb with host cells rather than by any direct effect on bacterial physiology. We conclude that uncontrolled intracellular growth of Mtb in DS LoCs is largely attributable to reduced surfactant secretion.

## Attenuation of an ESX-1-deficient strain of Mtb is independent of surfactant

We examined whether Mtb mutants that were previously shown to be attenuated in the mouse model of TB are also attenuated in the LoC model and whether surfactant plays a role in attenuation. Mtb lacking both the isocitrate lyase genes *icl1* and *icl2* grows normally under standard conditions in vitro but is incapable of growth in the lungs of mice and is rapidly cleared (*Muñoz-Elías and McKinney, 2005*). In the LoC model, we found that the Δ*icl1*Δ*icl2* strain is unable to grow in either ATs or macrophages even under the more-permissive DS conditions (*Figure 3C vs. A*, *Figure 3—figure supplement 1A–C* for widefield images and *Figure 3—figure supplement 1C–F* for confocal images), indicating that attenuation of this mutant is similar in the LoC and mouse models and independent of surfactant secretion. These results once again are suggestive of intracellular Mtb growth in the LoC model because the Δ*icl1*Δ*icl2* strain has a growth defect relative to wild-type only when intracellular in a host cell due to the accumulation of metabolic intermediates (*Upton and McKinney, 2007*), whereas there are no differences in growth between the axenic cultures of Δ*icl1*Δ*icl2* and wild-type strains in 7H9 medium (*Muñoz-Elías and McKinney, 2005*) or in the ALI media used in the LoC experiments (*Figure 2—figure supplement 3*).

The activity of the ESX-1 Type VII secretion system, a major Mtb virulence factor that is required for escape from the phagosome into the cytosol (*van der Wel et al., 2007*), is upregulated during AT infection (*Ryndak et al., 2015*). In comparison to wild-type Mtb, whose intracellular growth rate is strongly dependent

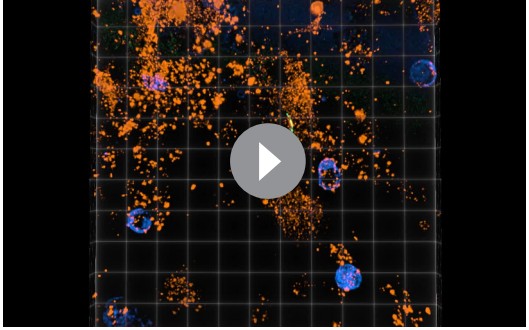

**Video 2.** 3D rotation along the Y-axis of a Z-stack highlighting the intracellular localization of Mtb within an infected AT on an LoC at 4 days post-infection. The field of view is 40.30 × 40.30 × 23.40 μm$^3$. Colors: Actin is labeled with the Amber LUT , Mtb with the Spring Green LUT, macrophages (identified via GFP expression are labeled with the Azure LUT), and nuclei stained by DAPI is labeled with the Electric Indigo LUT.

https://elifesciences.org/articles/59961#video2

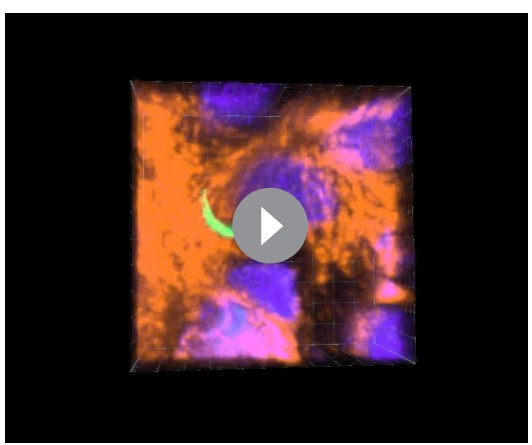

**Video 3.** Dynamic animation of the Z-stack highlighting the intracellular localization of Mtb within an infected AT on an LoC at 4 days post-infection. The field of view is 40.30 × 40.30 × 23.40 μm³. Colors: Actin is labeled with the Amber LUT, Mtb with the Spring Green LUT, macrophages (identified via GFP expression are labeled with the Azure LUT), and nuclei stained by DAPI is labeled with the Electric Indigo LUT.
https://elifesciences.org/articles/59961#video3

on surfactant levels (*Figure 3D–G*), intracellular growth of the 5'Tn::*pe35* strain (*Chen et al., 2013*) that is deficient in ESX-1 secretion but retains PDIM secretion (*Figure 4—figure supplement 1*) is largely independent of surfactant levels (*Figure 3H–K*). Under NS conditions, a greater fraction of ESX-1-deficient bacteria are 'non-growing' in ATs (*Figure 3I*). Under DS conditions, the ESX-1-deficient strain is unable to grow as rapidly as wild-type in both macrophages and ATs (*Figure 4B,D*) and a fraction (ca. 12%) of ESX-1-deficient bacteria are non-growing in macrophages (*Figure 3K*). This attenuation in DS conditions is evident by visual inspection at 6 dpi (*Figure 3B* vs 3A). Macrophages are less permissive than ATs for intracellular growth of ESX-1-deficient Mtb under both NS and DS conditions (*Figure 3—figure supplement 2C,D*). In contrast, wild-type Mtb, grows more slowly in macrophages than in ATs only under NS *but not* DS conditions (*Figure 3—figure supplement 2A,B*). Overall, attenuation of ESX-1-deficient bacteria relative to wild-type is not rescued by surfactant deficiency. This demonstrates that ESX-1 secretion is necessary for rapid intracellular growth in the absence of surfactant, consistent with the hypothesis that surfactant may attenuate Mtb growth by depleting ESX-1 components on the bacterial cell surface (*Raffetseder et al., 2019*).

## LoC model in NS conditions accurately reflects in vivo Mtb growth dynamics

Although median values of growth rate per hour are similar for wild-type and strains of Mtb under NS conditions (*Figure 4A,C* vs. *Figure 4B,D*), subtle differences in the probability density functions for each distribution (reflected in the 1–99 percentile interval in *Figure 4A,C*) could nevertheless generate significant differences in population sizes over a few days. This is particularly true for TB where growth is exponential in early infection, and bacterial numbers are enumerated over weeks or months of infection in the mouse model. To determine if this could account for the attenuation of the *esx-1* strain observed in vivo, we simulated the progression of a low-dose mouse infection

**Table 2.** Data for mean and median growth rates and total number of microcolonies (n) analyzed in the different experimental conditions outlined in *Figures 2* and *3*.

| Strain | Infection | Surfactant | Mean(h⁻¹) | Median (h⁻¹) | n |
|---|---|---|---|---|---|
| WT | AT | NS | 0.053 | 0.056 | 122 |
| WT | Mac | NS | 0.030 | 0.021 | 185 |
| WT | AT | DS | 0.111 | 0.103 | 219 |
| WT | Mac | DS | 0.109 | 0.107 | 122 |
| *esx-1* | AT | NS | 0.054 | 0.062 | 61 |
| *esx-1* | Mac | NS | 0.027 | 0.032 | 93 |
| *esx-1* | AT | DS | 0.073 | 0.081 | 25 |
| *esx-1* | Mac | DS | 0.045 | 0.047 | 55 |
| WT | Mac | DS-chiptreat | 0.073 | 0.079 | 121 |
| WT | Mac | DS-Mtbtreat | 0.055 | 0.061 | 63 |

(infectious dose = 50 CFU at 1 dpi) using intracellular bacterial growth rates randomly chosen from the growth rate distributions of each strain in macrophages in the LoC model (*Figure 3F,J*). At 2 dpi, population sizes for the ESX-1-deficient strains (*Figure 4E*, n = 100, p=2.3×10$^{-6}$) are already significantly smaller than for wild-type Mtb. The levels of attenuation predicted by this simple model using values from NS (*Figure 4F*) *but not* DS (*Figure 4G*) LoC conditions are in good agreement

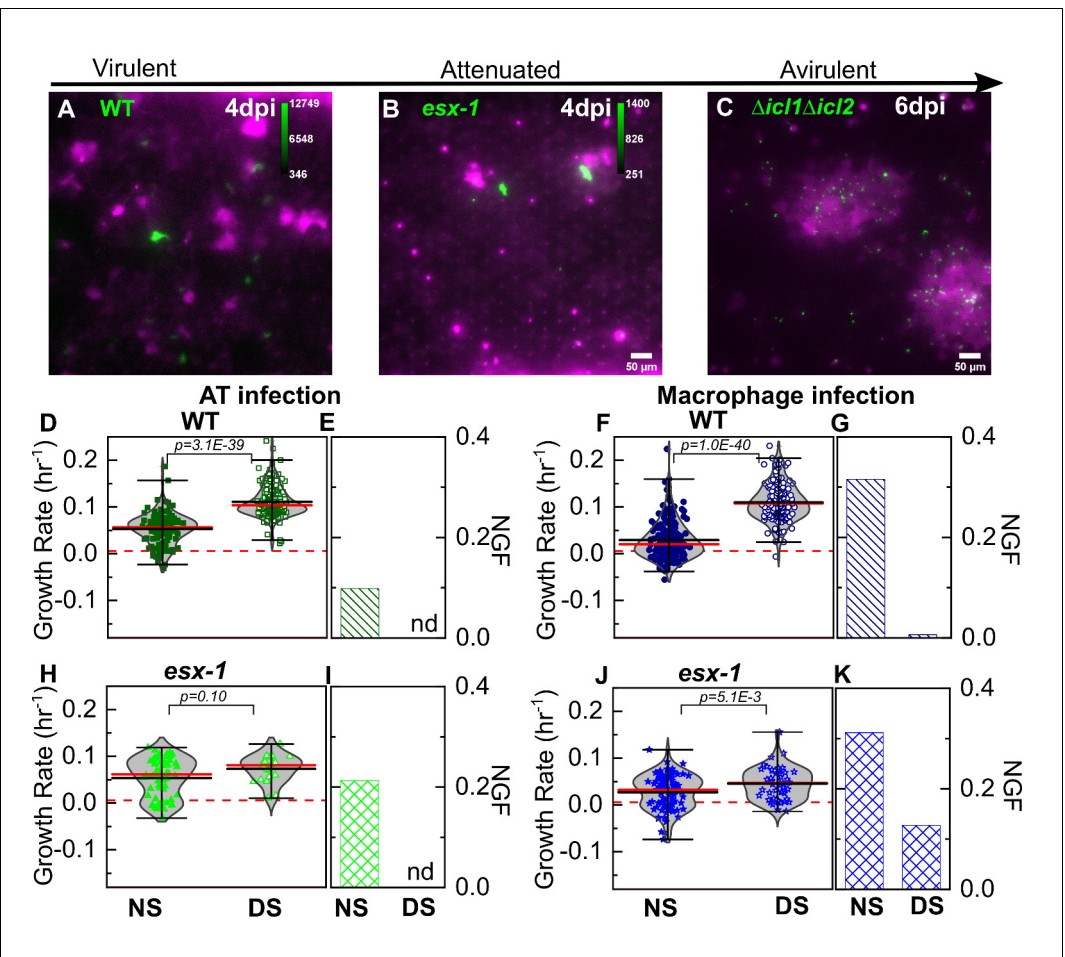

**Figure 3.** Attenuation of ESX-1-deficient strain is surfactant independent. Snapshots from live-cell imaging in deficient surfactant (DS) conditions at 4 days post-infection for (**A**) wild-type Mtb, (**B**) an attenuated ESX-1-deficient strain (*esx-1*), and (**C**) at 6 days post-infection for the avirulent Δicl1Δicl2 strain infected at a higher inoculum. The calibration bar (inset in **A-C**) indicates the corresponding absolute intensities in the Mtb channel for a direct comparison across strains. Growth of the ESX-1-deficient strain is attenuated relative to wild-type Mtb (~10-fold decrease in intensity) and the Δicl1Δicl2 strain is unable to grow. Scatter plots indicate Mtb growth rate in individual ATs (**D, H**) and macrophages (**F, J**) and bar graphs indicate the non-growing fraction (NGF) of bacteria for infected ATs (**E, I**) and macrophages (**G, K**) in normal surfactant (NS) and DS conditions for wild-type and ESX-1-deficient strains. 'nd', not detected. Each dataset was fitted with a non-parametric kernel density estimation characterized by the mean (black) and median (red) values and whiskers represent the 1-99 percentile interval. For wild-type Mtb, DS conditions significantly increase growth rates and lower the NGF in both ATs (**D, E**) (n=122 for NS, n=219 for DS, p=3.1E-39) and macrophages (**F, G**) (n=185 for NS, n=122 for DS, p=1.0E-40). For the ESX-1-deficient strain, differences between NS and DS conditions are not significant in ATs (**E**) (n=61 for NS, n=25 for DS, p=0.10); for macrophage infection, differences are statistically significant (**J**) (n=93 for NS, n=55 for DS, p=5.1E-3), but a significant number of bacteria remain non-growing in DS conditions (**K**).

The online version of this article includes the following figure supplement(s) for figure 3:

**Figure supplement 1.** Additional characterization of a DS LoC infected with the Δicl1Δicl2 Mtb strain at 6 dpi.

**Figure supplement 2.** Additional characterization of the Mtb growth rates in ATs vs. macrophages under DS and NS conditions.

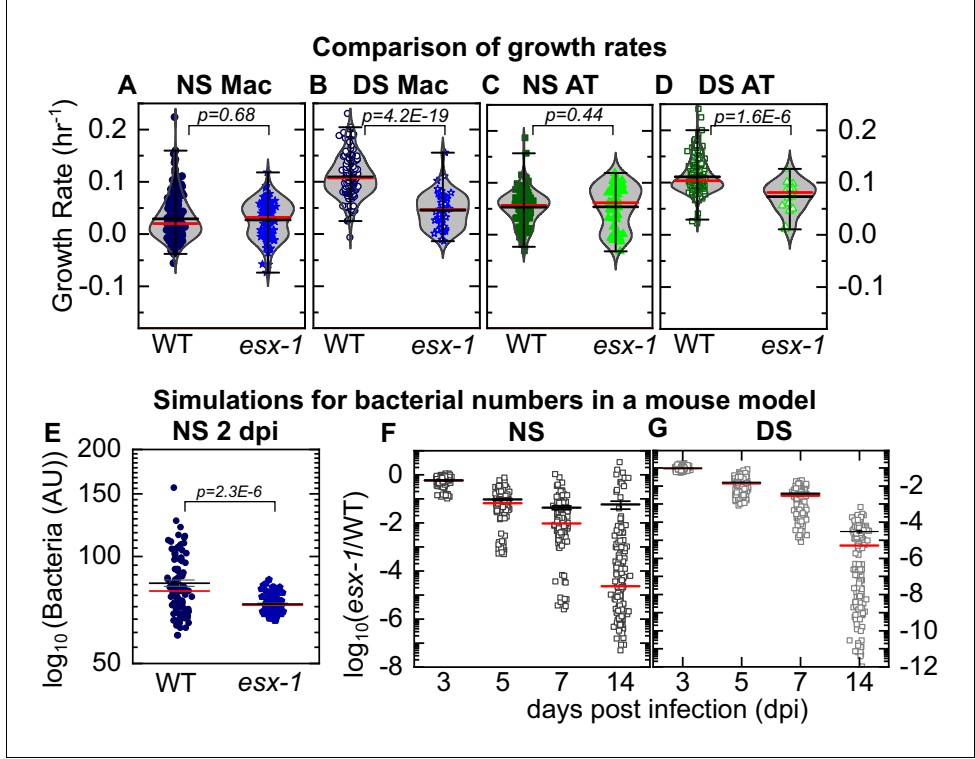

**Figure 4.** LoC model under normal surfactant (NS) conditions accurately reflects in vivo dynamics. Experimental data: Scatter plots of growth rates of intracellular bacteria from *Figure 3* compared between infected macrophages (A, B) and ATs (C,D) under NS conditions (A, C) and deficient surfactant (DS) conditions (B, D) for wild-type and ESX-1-deficient strains (*esx-1*) of Mtb. The number of samples for each bacterial strain and LoC condition is given in *Table 2*. Mean and median values are indicated by black and red lines, respectively, and whiskers indicate the 1–99 percentile interval. p-Values were calculated using the Kruskal-Wallis one-way ANOVA test. Simulations: Simulations of independent low-dose aerosol infections with 50 WT or *esx-1* bacteria. Simulated bacteria grow at rates randomly chosen from the kernel density estimations for the respective populations in *Figure 3F and J*, respectively. (E) In NS conditions, mean bacterial numbers for wild-type Mtb are significantly higher (p = 2.3E-6, n = 100) than for ESX-1-deficient Mtb. (F, G) Plots of the logarithm of ESX-1-deficient population size relative to WT (*esx-1*/WT) at the indicated timepoints for NS (F) and DS (G) conditions. Each datapoint represents the mean (*esx-1*/WT) ratio from five mice bootstrapped from the larger population (n=1000) for each strain. The attenuation of the ESX-1-deficient strain initially increases but then levels off with a spread of (*esx-1*/WT) ratios by 14 days post-infection. Mean (black) and median (red) values are indicated, and whiskers indicate the standard error of the mean.

The online version of this article includes the following figure supplement(s) for figure 4:

**Figure supplement 1.** Characterization of the ESX-1 deficient strain for PDIM production.

with the experimental data for both mutants from the mouse model in the acute phase of infection (*Chen et al., 2013*; *Rao et al., 2005*). These results provide a strong validation that surfactant secretion by freshly isolated ATs in NS conditions in the LoC model provide a better mimic of the native lung environment than DS conditions and serve to benchmark the LoC model.

## Curosurf binds to the Mtb cell surface and removes virulence-associated lipids

Although attenuation of the ESX-1-deficient strain is completely independent of surfactant in ATs (*Figure 3H,I*), surfactant still has a small but significant impact on growth of ESX-1-deficient Mtb in macrophages (*Figure 3J,K*). We therefore hypothesized that an additional mechanism of surfactant-dependent protection could be the removal of virulence-associated lipids from the Mtb cell surface. Consistent with this idea, microscopic examination of Mtb exposed to fluorescently labeled Curosurf with a fluorescent analogue of DPPC (*Figure 5—figure supplement 1A*) revealed that surfactant

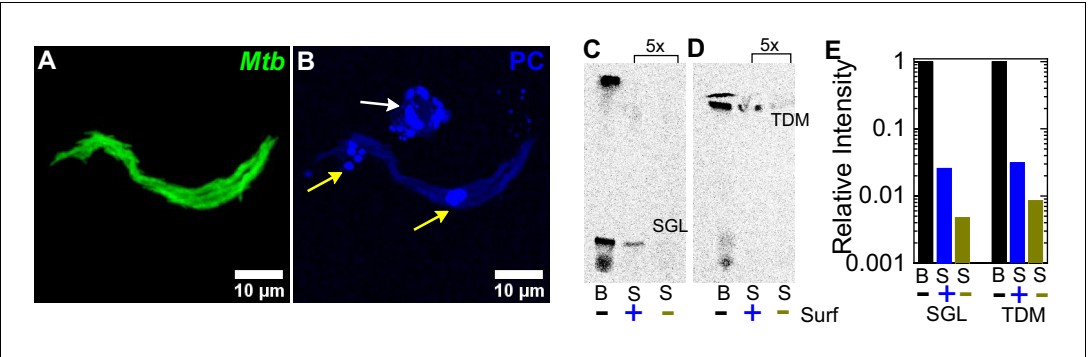

**Figure 5.** Surfactant depletes virulence factors on the bacterial cell surface. (**A, B**) Maximum intensity projections of an aggregate of fluorescent Mtb (false-colored green, (**A**)) after incubation with a 1% solution of Curosurf labeled with 10% v/v TopFluor-Phosphatidylcholine (PC) (false-colored blue, (**B**)). PC is incorporated into surfactant micelles (indicated by arrows in (**B**)), which also associate with and coat the bacteria (yellow arrows). (**C, D**) Thin-layer chromatography of total free lipid extracted from wild-type Mtb (**B**) or bacterial culture supernatants (S) with (+) or without (-) Curosurf pre-treatment. The latter two samples were spotted 5x in excess. Running solvents are (**C**) 90:10:1 chloroform: methanol: water to identify sulfoglycolipids (SGL), and (**D**) 80:20:2 chloroform: methanol: ammonium hydroxide to identify trehalose dimycolate (TDM). (**E**) Intensities of the SGL and TDM bands for the three samples in (**C, D**) are plotted relative to that for the bacterial sample without surfactant treatment (labeled 'B'). p-Values were calculated using the Kruskal-Wallis one-way ANOVA test.

The online version of this article includes the following figure supplement(s) for figure 5:

**Figure supplement 1.** Additional characterization of Curosurf-Mtb surface interactions.
**Figure supplement 2.** Curosurf treatment does not remove PDIM or TAG from the Mtb cell surface.

---

readily coated the bacteria (*Figure 5A,B*, *Figure 5—figure supplement 1B–E*), albeit heterogeneously (compare *Figure 5—figure supplement 1B,C* with *Figure 5—figure supplement 1D,E*). We also examined the effect of surfactant on the composition of the Mtb cell surface by comparing the total cell-associated lipids and the free (released) lipids prepared from untreated and Curosurf-treated Mtb. We found that Curosurf partially strips the Mtb cell surface of sulfoglycolipids (SGL) (*Figure 5C,E*) and trehalose dimycolate (TDM) (*Figure 5D,E*), but not phthiocerol dimycocerosates (PDIM) (*Figure 5—figure supplement 2*). The surfactant-mediated removal of these virulence-associated lipids (*Dulberger et al., 2020*) suggests an additional mechanism for the attenuation of intracellular growth of Mtb in macrophages.

## Discussion

ATs are the major cellular component of the distal lung, yet despite sporadic reports of AT infection in human TB (*Barrios-Payán et al., 2012*; *Hernández-Pando et al., 2000*), the role of ATs remains controversial. Previous work has revealed the specific roles of hydrophilic SP-A and SP-D proteins (*Ferguson et al., 2006*; *Pasula et al., 1997*) in altering the uptake and intracellular processing of Mtb in macrophages. Surfactant hydrolase enzymes within the alveolar lining fluid *Arcos et al., 2011*; *Scordo et al., 2017* have also been shown to release Mtb cell wall fragments as well as alter Mtb growth in ATs. Together, these studies show that, for the most part, specific components of surfactant can suppress intracellular Mtb growth. However, surfactant lipids have also been reported to upregulate Mtb growth via increased expression of CD36 (*Dodd et al., 2016*), and the alveolar lining fluid from some patients has also been shown to increase intracellular replication in epithelial cells and exacerbate infection (*Scordo et al., 2019*). However, all these studies focused on the role of specific components of surfactant and none of these studies probed the role of endogenous surfactant expression in co-cultures maintained at the ALI. In contrast, in the LoC model of early TB presented here, secretion of native surfactant by ATs at an ALI can be modulated. This physiological perturbation, which cannot be achieved in animal models due to the lethality of surfactant deficiency, provides a comprehensive view of the role of ATs in early TB and is an important advance over previous co-culture models for TB (*Bermudez et al., 2002*; *Bielecka et al., 2017*; *Parasa et al.,*

*2014*). The unexpectedly rapid and uncontrolled intracellular growth of Mtb at the ALI in the absence of surfactant has not been reported previously in simpler in vitro models of host cell infection. Under NS conditions, we identified a substantial non-growing fraction of intracellular Mtb in both ATs and macrophages, which may be equivalent to the 'non-growing but metabolically active' (NGMA) populations previously observed in the lungs of mice (*Manina et al., 2015*). These examples underscore some of the advantages of the LoC model, which provides a more faithful mimesis of the complex in vivo environment compared to conventional in vitro models.

Taken together, our findings indicate that pulmonary surfactant plays an important role in host innate immunity during early Mtb infection, which may partly explain why individuals with defective surfactant function (*Finley and Ladman, 1972*; *Subramaniam et al., 1995*; *Moliva et al., 2019*) also show an increased risk of developing active TB. We also report that ATs are more permissive to Mtb growth than macrophages under NS (but not DS) conditions. Infection of ATs is not an artefact of the LoC model, as we also identified infected ATs in the lungs of aerosol-infected mice using a sensitive microscopy-based approach, which might be more discriminating than FACS-based approaches used previously (*Cohen et al., 2018*). Alveolar macrophages in the lung have been shown to be sessile (*Westphalen et al., 2014*), and so the higher incidence of AT infection in the LoC model is probably due to differences in the method of Mtb inoculation. This in turn suggests a deeper link between the airway and alveolar geometries that determine aerosol deposition in the lungs (which are not captured in the LoC model but could be addressed with 3D bioprinted models [*Grigoryan et al., 2019*]) and resident lung immunity. Given the very low inoculum size in human Mtb (most infected individuals harbor just one primary lesion originating from a single bacillus *Mckinney et al., 1998*), we speculate that first contact with an AT, albeit much rarer in vivo could potentially lead to a more aggressive infection. This could provide one explanation for the observation that the proportion of exposed individuals who develop clinical TB is low.

Time-lapse imaging at an ALI in the LoC infection model directly quantifies bacterial growth rates with a spatiotemporal resolution that is unachievable using indirect measurements of bacterial growth rates in mouse models of TB (e.g. plating tissue homogenates to measure colony-forming units [CFU]). The deliberate choice to use murine over human cells allows us to add macrophages to the chip from a mouse line that constitutively expresses GFP, and provides unambiguous identification of these cells over multiple days which would not be possible with CellTracker and other fluorescent dyes. It also enables us to benchmark the model with previous reports from the mouse model for TB. For example, average growth rates for wild-type Mtb under NS conditions are in good agreement with net Mtb growth rates measured using a plasmid-loss assay in mice (*Gill et al., 2009*). However, our microscopy-based approach also reveals the population distribution of growth rates, which highlights that even under NS conditions, a small proportion of cells show robust intracellular Mtb growth, and that robust and attenuated Mtb growth can occur in close proximity to each other on-chip. These observations reinforce an important role for cell-to-cell heterogeneity in host-Mtb outcomes (*Lin et al., 2014*). The growth characteristics of ESX-1-deficient Mtb in our LoC model are also in agreement with experiments from animal models, further validating the LoC model.

Exogenous addition of Curosurf, a surfactant replacement formulation of phospholipids and hydrophobic proteins, rescues the effects of surfactant deficiency on mycobacterial growth to a large extent. Phospholipids are the main component of native surfactant, and lipid recycling is a key function of type II ATs and alveolar macrophages (*Olmeda et al., 2017*). The two-way interaction of surfactant with the bacterial cell surface (surfactant removes virulence-associated Mtb surface lipids and proteins, and surfactant lipids coat the Mtb surface) may alter how host cells take up these bacteria, or how they are processed after uptake. More generally, surfactant phospholipids have been shown to have antiviral properties and can serve as a potent adjuvant for antiviral vaccines either on their own (*Kimoto et al., 2013*) or as a medium for delivery of immune-stimulating molecules to ATs (*Wang et al., 2020*). Our findings suggest a potential role for pulmonary surfactant replacement formulations in host-directed therapies against TB. These insights were made possible by use of an organ-on-chip system that reproduces host physiology in a modular and tuneable fashion, which is frequently impossible to achieve in vivo.

# Materials and methods

## Key resources table

| Reagent type (species) or resource | Designation | Source or reference | Identifiers | Additional information |
|---|---|---|---|---|
| Biological sample (*Mus musculus*) | C57BL/6 primary alveolar epithelial cells | Cell Biologics | Cat#: C57-6053 | |
| Biological sample (*Mus musculus*) | C57BL/6 primary lung microvascular endothelial cells | Cell Biologics | Cat#: C57-6011 | |
| Other | Epithelial cell culture medium with kit | Cell Biologics | Cat#: M6621 | |
| Other | Endothelial cell culture medium with kit | Cell Biologics | Cat#: M1168 | |
| Strain, strain background (*Mus musculus, C57BL6*) | Tg(act-EGFP) 131Osb/LeySopJ | Jackson Laboratory | Cat#: 006567; RRID:IMSR_JAX:006567 | Female, 6–8 weeks old |
| Peptide, recombinant protein | Murine recombinant M-CSF | ThermoFisherScientific | Cat#: PMC2044 | |
| Commercial assay or kit | Superscript IV First Strand Synthesis System | Thermo Fisher Scientific | Cat#: 18091050 | |
| Commercial assay or kit | Sybr Green PCR Master Mix | Thermo Fisher Scientific | Cat#: 4334973 | |
| Antibody | Anti-mouse proSPC (Rabbit polyclonal) | Abcam | Cat#: ab40879; RRID:AB_777473 | IF(1:100) |
| Antibody | Anti-mouse Podoplanin-488 (Syrian Hamster monoclonal) | Thermo Fisher Scientific | Cat#: 53-5381-82; RRID:AB_1106990 | IF(1:100) |
| Antibody | Anti-mouse CD45-647 (Rat monoclonal) | BioLegend | Cat#: 103124; RRID:AB_493533 | IF (1:100) |
| Commercial assay or kit | Lung dissociation kit - mouse | Miltenyi Biotec | Cat#: 130-095-927 | |
| Peptide, recombinant protein | Fibronectin from human plasma | Sigma-Aldrich | Cat#: F1056 | |
| Peptide recombinant protein | Native Collagen, Bovine dermis | AteloCell | Cat#: IAC-50 | 5 mg/ml |
| Software, algorithm | FIJI | | RRID:SCR_002285 | |
| Software, algorithm | MATLAB | | RRID:SCR_001622 | |

## Cell culture

Primary C57BL/6 alveolar epithelial cells (ATs) and lung microvascular endothelial cells were obtained from Cell Biologics, USA, and were certified mycoplasma negative by the supplier. Each vial of ATs consisted of a mix of Type I and Type II ATs, which was verified by both immunostaining and qRT-PCR for type I and type II markers (*Figure 1A–C*, *Figure 1—figure supplement 1*). Both cell types were cultured in vitro in complete medium comprising base medium and supplements (Cell Biologics, USA) in 5% $CO_2$ at 37°C. NS ATs were seeded directly on the LoC (see below), without prior in vitro culture. DS ATs were passaged 6–11 times before use and were verified to be free of mycoplasma contamination prior to use.

## Bone marrow isolation and culture

Bone marrow was obtained from 6- to 8-week-old Tg(CAG-EGFP)131Osb/LeySopJ (also known as Tg(act-EGFP) Y01Osb) mice (Jackson Laboratories, USA, Stock Number 006567) and cryopreserved. This transgenic line constitutively expresses enhanced GFP under the control of the chicken beta-actin promoter and the cytomegalovirus enhancer. Mice were housed in a specific pathogen-free

facility. Animal protocols were reviewed and approved by EPFL's Chief Veterinarian, by the Service de la Consommation et des Affaires Vétérinaires of the Canton of Vaud, and by the Swiss Office Vétérinaire Fédéral. Bone marrow was cultured in Dulbecco's Modified Eagle Medium (DMEM) (Gibco) supplemented with 10% fetal bovine serum (FBS, Gibco) and differentiated for 7 days with 20 ng/ml recombinant murine Macrophage-Colony Stimulating Factor protein (M-CSF) (Thermo Fisher Scientific). Bone marrow was cultured in plastic petri dishes without pre-sterilization (Greiner Bio-one) so that differentiated macrophages could be detached. No antibiotics were used in the cell culture media for all cell types to avoid activation of macrophages or inhibition of Mtb growth, and the frozen marrow was verified to be free of mycoplasma contamination prior to use.

## RNA isolation from NS or DS ATs and cells from NS or DS LoCs

Freshly isolated ATs (NS) were grown overnight in cell-culture microdishes (Ibidi) or T-25 cell culture flask (TPP, Switzerland). Passaged ATs (DS) were grown to confluency in a T-75 cell culture flask (TPP). Growth media was removed from the flask, and the cells were incubated with the appropriate volume of TRIzol (Ambion) as per the manufacturer's instruction. TRIzol-treated cell lysates were stored at −20°C before further processing. RNA was precipitated with isopropanol, washed in 75% ethanol, resuspended in 50 µl of DEPC-treated water, treated with Turbo DNase (Ambion), and stored at −80°C until use. DNase-treated RNA was used to generate cDNA using the SuperScriptII First-Strand Synthesis System with random hexamers (Invitrogen), and was stored at −20°C.

For RNA isolation from LoCs, one NS or DS LoC each were established as per the protocols described and maintained for 24 hr at the ALI. RNA from the apical and vascular channels was isolated separately in approximately 350 µl of the RLT Plus buffer of the Qiagen Micro Plus RNA Isolation Kit, and subsequently processed as per the manufacturer's instructions.

cDNA was generated using the SuperScript IV First-Strand Synthesis System with random hexamers (Invitrogen), and subsequently stored at −20°C.

## Quantitative real-time PCR (qRT-PCR)

Specific primers used are listed in *Table 3*. Sequences for the primers for *Pdpn*, *Cav1*, and *Igfbp2* were obtained from *Wang et al., 2018* and the sequences for the remaining primers were obtained from Origene and primers were obtained from a commercial supplier (Microsynth, Switzerland). qRT-PCR reactions were prepared with SYBRGreen PCR Master Mix (Applied Biosystems) with 1 µM primers, and 1 or 2 µl cDNA. Reactions were run as absolute quantification on ABI PRISM7900HT Sequence Detection System (Applied Biosystems). Amplicon specificity was confirmed by melting-curve analysis.

## AT characterization via immunofluorescence

Freshly isolated ATs (NS) or passaged ATs (DS) were grown overnight in 35-mm cell-culture microdishes (Ibidi GmbH, Germany). The confluent layer of cells was subsequently fixed with 2% paraformaldehyde (Thermo Fisher Scientific) in phosphate-buffered saline (PBS, Gibco) at room temperature for 30 min, washed with PBS, and incubated with a blocking solution of 2% bovine serum albumin (BSA) in PBS for 1 hr at room temperature. The blocking solution was removed, and the cells were incubated with the primary antibody (1:100 dilution in 2% BSA solution in PBS) overnight at 4°C. Antibodies used were anti-Podoplanin Monoclonal Antibody (eBio8.1.1 (8.1.1)), Alexa Fluor 488, eBioscience (ThermoFisher Scientific), and anti-pro-SPC antibody (ab40879, Abcam). The cell-culture microdishes were washed 3x in PBS, incubated with a fluorescent secondary antibody (Donkey anti-rabbit Alexa Fluor 568 (A10042 Thermo Fisher)) in a solution of 2% BSA in PBS for 1 hr at room temperature, then thoroughly washed with PBS and incubated with Hoechst 33342 nuclear staining dye (1:1000 dilution, Thermo Fisher Scientific) for 15–20 min for nuclear staining. Confocal images were obtained on a Leica SP8 microscope in the inverted optical configuration at the EPFL BIOP core facility.

## Bacterial culture

All bacterial strains were derived from Mtb strain Erdman and cultured at 37°C. Liquid medium: Middlebrook 7H9 (Difco) supplemented with 0.5% albumin, 0.2% glucose, 0.085% NaCl, 0.5% glycerol, and 0.02% Tyloxapol. Solid medium: Middlebrook 7H11 (Difco) supplemented with 10% OADC

**Table 3.** Primers used for qPCR characterization of gene expression of the NS and DS AT cells in *Figure 1*, *Figure 1—figure supplement 1*, .

| qPCR primer list |
| --- |
| 5'- CATCACTGCCACCCAGAAGACTG-3' *Gapdh* forward |
| 5'- ATGCCAGTGAGCTTCCCGTTCAG-3' *Gapdh* reverse |
| 5'- ACCTGGATGAGGAGCTTCAGAC-3' *Sftpa* forward |
| 5'- CTGACTGCCCATTGGTGGAAAAG-3' *Sftpa* reverse |
| 5'- TGTCCTCCGATGTTCCACTGAG-3' *Sftpb* forward |
| 5'- AGCCTGTTCACTGGTGTTCCAG-3' *Sftpb* reverse |
| 5'- GTCCTCGTTGTCGTGGTGATTG-3' *Sftpc* forward |
| 5'- AAGGTAGCGATGGTGTCTGCTC-3' *Sftpc* reverse |
| 5'- AGGTCCAGTTGGACCCAAAGGA-3' *Sftpd* forward |
| 5'- CTGGTTTGCCTTGAGGTCCTATG-3' *Sftpd* reverse |
| 5'- CTTCATGGACGAAGCTGACCTG-3'*Abca3* forward |
| 5'- GTGCGGTTCTTTTACCAGCGTC-3' *Abca3* reverse |
| 5'-TCCATGAACCCAGCCCGATCTT-3'; *Aqp5* forward |
| 5'-GAAGTAGAGGATTGCAGCCAGG-3'; *Aqp5* reverse |
| 5'- CAAGAAAACAAGTCACCCCAATAG-3'; *Pdpn* forward |
| 5'- AACAATGAAGATCCCTCCGAC-3'; *Pdpn* reverse |
| 5'-CGAGGTGACTGAGAAGCAAG-3'; *Cav1* forward |
| 5'-TCCCTTCTGGTTCTGCAATC-3'; *Cav1* reverse |
| 5'-TGCCAAACACCTCAGTCTG-3'; *Igfbp2* forward |
| 5'-AGGGAGTAGAGATGTTCCAGG-3'; *Igfbp2* reverse |
| 5'-GGTGATATTCGAGACCATTTACTG-3'; *Cxcl15* forward |
| 5'-GCCAACAGTAGCCTTCACCCAT-3'; *Cxcl15* reverse |

enrichment (Becton Dickinson) and 0.5% glycerol. Aliquots were stored in 15% glycerol at −80°C and used once. All strains were transformed with a plasmid integrated at the chromosomal *attB* site to allow constitutive expression of the fluorescent protein tdTomato under the control of the hsp60 promoter. Wild-type (WT) refers to the Erdman strain constitutively expressing tdTomato. The 5'Tn:: *pe35* (ESX-1 deficient) strain was generated using transposon mutagenesis (*Chen et al., 2013*).

## Infection of mice with Mtb

Female C57BL/6 mice (Charles River Laboratories) were housed in a specific pathogen-free facility. Animal protocols were reviewed and approved by EPFL's Chief Veterinarian, by the Service de la Consommation et des Affaires Vétérinaires of the Canton of Vaud, and by the Swiss Office Vétérinaire Fédéral. Mice were infected by the aerosol route using a custom-built aerosol machine, as described (*MacMicking et al., 2003*). Bacteria were grown to exponential phase, corresponding to an optical density at 600 nm ($OD_{600}$) of 0.5, collected by centrifugation at 2850 *g* for 10 min, and resuspended in PBS supplemented with 0.05% Tween 80 (PBS-T). The bacterial suspension was subjected to low-speed centrifugation (700 *g*) for 5 min to remove bacterial aggregates. The cell suspension was adjusted to $OD_{600}$ 0.1 with PBS-T in a final volume of 20 ml, which was used to infect mice by aerosol. At 1 dpi, a group of four mice were euthanized by $CO_2$ overdose; the lungs were removed aseptically and homogenized in 3 ml of 7H9 medium. Serial dilutions were plated on 7H11 plates containing 100 μg/ml cycloheximide (Sigma), and colonies were counted after 4–5 weeks of incubation at 37°C. The aerosol infection corresponded to a bacterial load of between 60 and 100 CFU per mouse at 1 dpi.

## Extraction and characterization of a single-cell suspension of lung cells from Mtb-infected mice

At 8 dpi, a group of five mice were euthanized by an overdose of ketamine/xylazine anesthetic, and the lungs were washed with PBS delivered via injection through the right ventricle of the heart to remove excess red blood cells. Lungs from each mouse were removed aseptically, minced into small pieces with scissors, and added to 2.5 ml of lung dissociation media reconstituted as per the manufacturer's instructions (Lung Dissociation Kit – Mouse, Miltenyi Biotec). The lungs were then dissociated using a gentleMACS Octo Dissociator (Miltenyi Biotec). The resulting homogenate was filtered through a 40-µm cell filter, centrifuged for 10 min at 300 $g$, and resuspended in alveolar epithelial cell media supplemented with 10% FBS. The homogenate was then plated in 50-mm glass-bottom cell-culture dishes (Ibidi) and incubated for 36–48 hr to allow for epithelial cell attachment. Additional medium was added to each cell-culture dish at 24 hr.

## Quantification of Mtb-infected ATs within the adherent cell fraction

Adherent cells from the single-cell suspension were subsequently fixed with paraformaldehyde and stained for immunofluorescence as already described. Antibodies: anti-Podoplanin Monoclonal Antibody (eBio8.1.1 (8.1.1)), Alexa Fluor 488, eBioscience (ThermoFisher Scientific) to label Type I ATs, anti-pro-SPC antibody (ab40879, Abcam) followed by secondary antibody staining (Donkey anti rabbit Alexa Fluor 488 (A21206 Thermo Fisher)) to label Type II ATs, and Alexa 647 anti-CD45 antibody (103124, BioLegend) to label immune cells.

## Murine LoC model

LoCs made of polydimethylsiloxane (PDMS) were obtained from Emulate (Boston, USA). Extracellular matrix (ECM) coating was performed as per the manufacturer's instructions. Chips were activated using ER-1 solution (Emulate) dissolved in ER-2 solution at 0.5 mg/ml (Emulate) and exposed for 20 min under UV light. The chip was then rinsed with coating solution and exposed again to UV light for a further 20 min. Chips were then washed thoroughly with PBS before incubating with an ECM solution of 150 µg/ml bovine collagen type I (AteloCell, Japan) and 30 µg/ml fibronectin from human plasma (Sigma-Aldrich) in PBS buffered with 15 mM HEPES solution (Gibco) for 1–2 hr at 37°C. If not used directly, coated chips were stored at 4°C and pre-activated before use by incubation for 30 min with the same ECM solution at 37°C. Endothelial cells were cultured overnight at 37°C and 5% $CO_2$ in T-75 cell culture flasks, detached with 0.05% Trypsin, concentrated to 5–10 million cells/ml, and seeded on the bottom face of the PDMS membrane. The chip was then incubated for a short period at 37°C to allow the endothelial cells to spread and subsequently seeded with ATs. Freshly isolated ATs were seeded directly from cryopreserved vials received from the supplier, whereas DS LoCs were seeded from cells cultured overnight at 37°C and 5% $CO_2$, in both cases at a concentration of 1–2 million cells/ml. The chip was incubated overnight with complete epithelial and endothelial media in the epithelial and endothelial channels, respectively, under static conditions. The next day, the chip was washed and a reduced medium for the ALI was flowed through the vascular channel using syringe pumps (Aladdin-220, Word Precision Instruments) at 60 µl/hr as described (*Hassell et al., 2017*). The composition of the ALI media used was as described in *Hassell et al., 2017* but with an FBS concentration of 5%. The epithelial face was incubated with epithelial base medium with 200 nM dexamethasone (Sigma Aldrich) without FBS supplementation to promote tight junction formation and surfactant expression as reported in previous LoC studies (*Huh et al., 2010*; *Hassell et al., 2017*). Flow was maintained over 2–3 days with daily replacement of the medium on the epithelial face (with dexamethasone supplementation). At the end of this period, GFP-expressing macrophages differentiated for 7 days in M-CSF (described above) were detached from the petri dish using 2 mM ethylenediaminetetraacetic acid (EDTA, Sigma Aldrich) in PBS at 4°C, centrifuged at 300 $g$ for 5 min, and resuspended in a small volume of epithelial cell media without dexamethasone. This solution containing macrophages was introduced onto the epithelial face and incubated for 30 min at 37°C and 5% $CO_2$ to allow macrophages to attach to the epithelial cells. Medium on the epithelial face was then removed and the chip was maintained overnight at the ALI. Chips that successfully maintained the ALI overnight were transferred to the biosafety level 3 (BSL-3) facility for Mtb infection. No antibiotics were used in any of the cell culture media for setting up the LoC model.

## Immunostaining of uninfected LoCs

Uninfected LoCs were maintained at an ALI for up to 7 days after addition of macrophages, during which time ALI medium was flowed through the endothelial channel at 60 µl/hr. After 7 days at the ALI, the chip was fixed for immunostaining as described above; a permeabilization step with a solution containing 2% w/v saponin (Sigma Aldrich) and 0.1% Triton X-100 (Sigma Aldrich) was performed before incubation with the secondary antibody. F-actin on both the epithelial and endothelial face was stained using Sir-Actin dye (Spherochrome) at 1 µM for 30 min concurrently with Hoechst staining, as described above. Confocal images were obtained on a Leica SP8 microscope in the inverted optical configuration at the EPFL BIOP core facility.

## Infection of the LoC with Mtb

The chip was assembled into a stage top incubator (Okolab, Italy) prior to infection and flow of medium through the vascular channel was maintained throughout the course of the experiment by use of a syringe pump. A 1 ml aliquot of a culture of Mtb grown to exponential phase ($OD_{600}$0.3–0.5) was centrifuged at 5000 $g$ for 5 min at room temperature, the supernatant was removed, and the cell pellet was resuspended in 200 µl of epithelial cell media without FBS. A single-cell suspension was generated via filtration through a 5-µm syringe filter (Millipore). The single-cell suspension was diluted 100-fold in epithelial media and 30 µl was added to the epithelial channel of the LoC. The infectious dose was measured by plating serial dilutions of the single-cell suspension on 7H11 plates and counting CFU after 3–4 weeks of incubation at 37°C and varied between 200 and 800 Mtb bacilli. The chip was incubated for 2–3 hr at 37°C and 5% $CO_2$ to allow Mtb infection of cells on the epithelial face, after which the solution on the epithelial face was withdrawn. The proportion of bacteria that remained on the chip was estimated by plating serial dilutions of the withdrawn solution on 7H11 plates and counting CFU after 3–4 weeks of incubation at 37°C. The epithelial face was returned to ALI and the inlets of the infected chip were sealed with solid pins as a safety precaution for time-lapse microscopy imaging in the BSL-3 facility.

## Time-lapse microscopy of the Mtb-infected LoC

The LoC was placed in a microscope stage-top incubator and mounted on the stage of a widefield Nikon Ti-2 microscope. The stage-top incubator was connected to a gas mixer (Okolab) to maintain 5% $CO_2$ throughout the imaging period. Flow of medium through the vascular channel was maintained throughout this period via the use of a syringe pump. The chip was imaged using a long working distance 20x phase-contrast objective (NA = 0.75, Ph2, Nikon) at 1.5 hr or 2 hr imaging intervals. The epithelial face of the chip (where the refractive index differences were highest due to the ALI) was maintained in focus using the Nikon Perfect Focus System. At each timepoint, a Z-stack of 9–10 images with an axial spacing of 10 µm was taken series for a series of fields of view along the length of the chip to account for the dynamic 3D movement of macrophages between both faces, as well as drift in focus over time. Each field of view was ~660 × 600 µm². Using a Sola SE II light source (Lumencor, USA), macrophages and Mtb were identified through fluorescence emission in the green (macrophages) and red (Mtb) channels using GFPHQ and mcherryHQ 32 mm dichroic filters, respectively. Phase-contrast images were also captured; the poor quality of these images due to the refractive index differences at the ALI serves as a continuous verification that ALI is maintained. All images were captured with an EMCCD camera (iXON Ultra 888, Andor) cooled to −65°C, with an EM gain setting of 300 to allow the sample to be illuminated with a low intensity of incident light in all fluorescent channels with reduced photodamage. Co-localization of the green and red fluorescence signals over a time course was identified as consistent with macrophage infection. Bacteria that did not co-localize with macrophages over time were assumed to infect ATs, which was verified by subsequent immunostaining.

## Data analysis of intracellular Mtb growth

Images were visualized using FIJI. Macrophage and AT infections from each field of view were visually curated by assessing the co-localization of fluorescent signals over time. Smaller stacks of one to two microcolonies were assembled. Custom-written software in MATLAB was used to measure the total fluorescence intensity of each intracellular bacterial microcolony which used the nestedSortStruct algorithm for MATLAB (*Hughley, 2018*; https://github.com/hugheylab/

nestedSortStruct) written by the Hughey lab. Briefly, at each timepoint, the Z-stack with the highest intensity in the fluorescence channel was identified; this image was then segmented to identify the bacterial microcolony; total fluorescence was measured by summing the intensity of all the pixels in this region after subtracting a value for each pixel that represented the average background fluorescence. We chose to measure the total fluorescence intensity because it accounts for both bacterial growth and dilution of the fluorescent protein due to growth (which is slow in a slow-growing bacterium like Mtb). We were unable to measure microcolony volumes accurately using widefield imaging due to poor axial resolution caused by large refractive index differences at the ALI; therefore, we obtained this value from only the Z stack with the highest intensity. Statistical analysis was performed using Origin 9.2 (OriginLabs), and p-values were calculated using the Kruskal-Wallis one-way ANOVA test, with the null hypothesis that the medians of each population were equal.

### Visualization of confocal images

Z-stacks from confocal images were visualized using ImageJ, 3D projection views and *Videos 1*, *2* and *3* were made using the ClearVolume plugin in ImageJ (*Royer et al., 2015*).

### Analysis of lamellar body volume

Custom written software in MATLAB was used to segment lamellar bodies in each slice of the Z-stack, measure area and intensity, and collate the mean intensity and volume over multiple slices, along the lines of the analysis of intracellular bacterial growth described above.

### Simulations of in vivo infections

Growth rate datasets for wild-type and ESX-1-deficient strains of Mtb in NS and DS LoC conditions were fitted with a non-parametric Kernel Smoothed distribution. We simulated a low-dose aerosol infection of 50 bacteria in the alveolar space of n = 100 or n = 1000 mice, and conservatively assumed that every bacterium interacted with a macrophage upon first contact. Each bacterium was assigned a growth rate picked at random from the Kernel Smoothed distributions and assumed to grow exponentially with these growth rates to generate an intracellular microcolony. The total bacterial numbers in each mouse at 2, 3, 5, 7, and 14 dpi were obtained by summing the bacterial counts from each microcolony for each mouse and are shown in *Figure 4E–G*. Total bacterial numbers for n = 100 mice of WT and ESX-1-deficient strains are shown in *Figure 4*.

### Curosurf treatment of DS LoCs

Curosurf (Chiesi Pharmaceuticals, Italy) was used as a 1% solution in epithelial medium for all LoC experiments. In the case where Curosurf was added to a DS LoC, a 1% solution was introduced to the epithelial face after the macrophages were added but before ALI was introduced for 2 min, and then removed. The following morning, this procedure was repeated just prior to the addition of the single-cell suspension of Mtb in the manner described above. Alternatively, a 1 ml aliquot of Mtb in exponential phase in 7H9 media was centrifuged at 5000 *g* for 5 min, resuspended in 1 ml of cell culture media containing 1% Curosurf, and incubated for 10–15 min at room temperature. This solution was then centrifuged again at 5000 *g* for 5 min and a single-cell suspension of Mtb was generated as described above. Fluorescent labeling of surfactant was achieved by adding TopFluor phosphatidylcholine (10% v/v, Avanti Polar Lipids) to Curosurf before dilution in cell culture medium.

### Total free lipid extraction and thin-liquid chromatography (TLC)

Mtb cultures (10 ml each) were grown to stationary phase in 7H9 with 10 $\mu C_i$ of $^{14}C$-propionate added during exponential phase. Total free lipid extraction from the bacterial pellet, supernatant, and supernatant from bacteria pre-treated with 3% Curosurf for 15 min at 37°C were extracted as described (*Parish and Roberts, 2015*). Extracted free lipids were air-dried, resuspended in 2:1 v/v solution of chloroform: methanol, and aliquots were spotted on 5 × 10 cm TLC silica gel 60 F$_{254}$ (Merck). Running solvent was 90:10:1 chloroform: methanol: water for the analysis of sulfoglycolipids (SGL), 80:20:2 chloroform: methanol: ammonium hydroxide for the analysis of TDM, and 9:1 petroleum ether: diethyl ether for the analysis of pthiocerol dimycocerosates (PDIM) and triacyclglycerols (TAG). The developed TLC plate was exposed to an Amersham Hyperfilm ECl (GE Healthcare) for

phosphorescence imaging and visualized with a Typhoon scanner (GE Healthsciences). Intensities of the bands observed were quantified using ImageJ.

For characterization of PDIM secretion by the WT and ESX-1-deficient strains, lipid extraction was performed via a protocol optimized for extraction of apolar lipids as described (*Ojha et al., 2008*). Extracted lipids were subsequently processed as described above.

## Additional information

### Competing interests
Riccardo Barrile: Riccardo Barrile was affiliated with Emulate Inc for a portion of the duration of this study. The author has no financial interests to declare. Katia Karalis: Katia Karalis is affiliated with Emulate Inc,. The author has no financial interests to declare. The other authors declare that no competing interests exist.

### Funding

| Funder | Grant reference number | Author |
| --- | --- | --- |
| Human Frontier Science Program | LT000231/2016-L | Vivek V Thacker |
| European Molecular Biology Organization | 921-2015 | Vivek V Thacker |
| Schweizerischer Nationalfonds zur Förderung der Wissenschaftlichen Forschung | 310030B_176397 | John D McKinney |

The funders had no role in study design, data collection and interpretation, or the decision to submit the work for publication.

### Author contributions
Vivek V Thacker, Conceptualization, Resources, Data curation, Software, Formal analysis, Funding acquisition, Validation, Investigation, Visualization, Methodology, Writing - original draft, Writing - review and editing; Neeraj Dhar, Conceptualization, Investigation, Methodology, Writing - review and editing; Kunal Sharma, Resources, Investigation, Methodology; Riccardo Barrile, Katia Karalis, Resources, Methodology; John D McKinney, Conceptualization, Supervision, Funding acquisition, Project administration, Writing - review and editing

### Author ORCIDs
Vivek V Thacker (iD) https://orcid.org/0000-0002-1681-627X
Neeraj Dhar (iD) http://orcid.org/0000-0002-5887-8137
Kunal Sharma (iD) https://orcid.org/0000-0001-8086-3436
Riccardo Barrile (iD) https://orcid.org/0000-0002-7301-3959
John D McKinney (iD) https://orcid.org/0000-0002-0557-3479

### Ethics
Animal experimentation: Animal protocols were reviewed and approved by EPFL's Chief Veterinarian, by the Service de la Consommation et des Affaires Vétérinaires of the Canton of Vaud, and by the Swiss Office Vétérinaire Fédéral (License Number VD 3434 for experiments involving organ collection and License Number VD 3472 for experiments involving infection with Mycobacterium tuberculosis).

### Decision letter and Author response
Decision letter https://doi.org/10.7554/eLife.59961.sa1
Author response https://doi.org/10.7554/eLife.59961.sa2

## Additional files

### Supplementary files

• Transparent reporting form

### Data availability

Figures in the main text include all the data for bacterial growth rates within the scatter plots, and all the data for qRT-PCR measurements and quantification of lamellar body size, number, and volume. A summary of the code used to calculate growth rates is included in the Materials and Methods. Annotated code written in Matlabused for data analysis for growth rates and for simulations of *in vivo* infections , raw data for bacterial fluorescence intensity over time that was used to calculate growth rates, and image stacks related to Fig. 1, Fig. 1 - figure supplement 3 and Fig. 2 - figure supplement 1 are available on Zenodo under https://doi.org/10.5281/zenodo.4266198.

The following dataset was generated:

| Author(s) | Year | Dataset title | Dataset URL | Database and Identifier |
|---|---|---|---|---|
| Thacker VV, Dhar N, Sharma K, Barrile R, Karalis K, McKinney JD | 2020 | A lung-on-chip model reveals an essential role for alveolar epithelial cells in controlling bacterial growth during early M. tuberculosis infection | https://doi.org/10.5281/zenodo.4266199 | Zenodo, 10.5281/zenodo.4266198 |

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
