## [Decision Letter]

**Acceptance summary:**

This article by Thacker and colleagues presents an organ-on-chip model to investigate the role of alveolar epithelial cells in controlling *Mycobacterium tuberculosis* infection via the production of surfactant. The authors found that the presence of surfactant is associated with a non-growing *M. tuberculosis* population within macrophages, whereas, there is more bacterial growth in the surfactant deficient condition. These are important findings to contribute to our understanding of early events during infection with this globally relevant pathogen as well as an interesting model to expand on in future studies.

**Decision letter after peer review:**

Thank you for submitting your article "A lung-on-chip model reveals an essential role for alveolar epithelial cells in controlling bacterial growth in early TB" for consideration by *eLife*. Your article has been reviewed by three peer reviewers, one of whom is a member of our Board of Reviewing Editors, and the evaluation has been overseen by Wendy Garrett as the Senior Editor. The following individual involved in review of your submission has agreed to reveal their identity: Michael U Shiloh (Reviewer #3).

The reviewers have discussed the reviews with one another and the Reviewing Editor has drafted this decision to help you prepare a revised submission.

Summary:

This paper by Thacker et al. describes the use of lung-on-a-chip microfluidic devices to study early interactions during *M. tuberculosis* infection under conditions meant to mimic the alveolar environment in vivo. The authors use time-lapse microscopy to study host cell-Mtb interactions in macrophages and alveolar epithelial cells and the impact of surfactant on Mtb infection. This study suggests that organ-on-a-chip systems might be able to reproduce elements of host-microbe physiology during infection, which is difficult to reproduce ex vivo using single cells, air-liquid interface, organoids or organ explants.

This is an exciting approach which has the potential to expand the ability to study host-pathogen interactions. However, the reviewers all agree that the manuscript requires a major revision and additional data. Specifically, the manuscript requires improvement in the cell identification/classification, co-localization of Mtb with epithelial cells and macrophages, and distinction between intracellular and extracellular growth in order for the authors to provide convincing data to support their interpretations and conclusions.

While the reviewers recognize that it is challenging to use live cell imaging in this system, much of the data of the paper, such as comparisons between infection of AECs and macrophages, rests on the ability to determine the precise localization of bacteria. However, neither AECs nor macrophages are specifically identified with high enough resolution to give confidence that the Mtb are associated with those cells specifically, and more importantly, that the bacteria are growing intracellularly rather than extracellularly. Many of the images are of such low resolution that only tiny dots of bacteria are observed.

In addition, the findings of attenuated growth of Mtb after exposure to surfactant in macrophages and alveolar epithelial cells, changes in the Mtb cell wall after exposure to surfactant, and the finding that exposure to surfactant does not alter the extracellular viability of *M. tuberculosis* have been reported by others using other in vitro models and should be discussed in manuscript.

Essential revisions:

1) Controls for the Model: What quality control is done for each experiment to determine the ratio of type I and type II AECs in each chip set up for each experiment? This is of particular importance because the authors do not show any images where they stain for both type I and type II AECs in the same chip. Do the authors have images stained for both type of cells to illustrate the composition of each chip? After Figure 1, what staining is done to confirm the DS cells decrease proSPC expression for each experiment? As this is a new model, better images showing well-defined cells and quantification of the number of alveolar epithelial cells type I and II, number of macrophages, numbers of endothelial cells, bacteria per cell, etc. would be helpful to evaluate the model. This applies particularly to Figures 1, 2 and Figure 3 supplement.

2) Cell Identification: The authors focus on the difference in surfactant gene expression in the newly isolated AECs (NS) versus in vitro passaged AECs (DS), but they also observe that aqp5 is downregulated. In fact, the data support that the cells are de-differentiating during passage in culture, which will have multiple effects on the cells, not just surfactant production. This needs to be further studied in detail to ensure that this cell is not an artifact produced by multi-passaging in vitro. After loss of those markers, how do the authors confirm they still have type I and type II AECs in their cultures? Is there microscopy data with other markers that are retained in the AECs? The authors should use several AT-IIs and AT-Is markers to be certain that the DS cell monolayers indeed still are AECs. Surfactant protein C, although used as a marker for AT-IIs, is a soluble protein that has been shown to interact with many cells within a cellular system. A correlation between SPTPC and AQP5 expression over time is also necessary as points out the differentiation of AT-IIs to AT-Is, a key feature of the role of AT-IIs as progenitors of AT-Is.

3) Analysis of Mtb Infection and Intracellular versus Extracellular Infection: The authors never stain for type I or type II AECs after infection and make the conclusion that the bacteria are within type II cells based on the absence of macrophage staining. However, the bacteria may not even be inside a cell, or the AECs could be dying during infection. On a related note, there are no data presented that show that type I cells are not infected in the lung on chip system with Mtb. Thus, higher resolution images, with clear colocalization and z-stacks, would increase the confidence in the results. The images and analysis provided do not distinguish between extracellular and intracellular growth. Mtb can form micro-colonies on the cell surface of alveolar epithelial cells and thus, the intracellular growth that they are reporting could be extracellular growth. The authors could treat the system with an antibiotic to kill extracellular Mtb attached to the alveolar epithelial cell surface. Also, can the authors discern between Mtb that are not growing vs. Mtb that are dead?

4) Studies with the Mtb Mutants: The data arguing for attenuation of Esx-1 mutant Mtb in AECs and macrophages are not strong, and the authors do not actually make a direct statistical comparison between appropriate groups (i.e. AEC NS WT vs Esx-1, or Mac NS WT vs Esx-1). For example, it appears that the mean/median growth rate of WT Mtb in macrophages is ~0.25hr-1, which appears roughly the same for Esx-1 mutant Mtb in the same cells. There may be a difference under DS conditions, but since the comparisons are not made directly it is impossible to know. The authors state that their data with the Esx1 mutant "demonstrates that ESX-1 secretion is necessary for rapid intracellular growth in the absence of surfactant, consistent with the hypothesis that surfactant may attenuate Mtb growth by depleting ESX-1 components on the bacterial cell surface". This seems like quite a jump in interpretation of the data since the Esx1 mutant is likely attenuated for many reasons, and this attenuation is dominant to any effect that surfactant is having. The authors also show that PDIM levels are not different in the presence or absence of surfactant, and this is an Esx1 dependent lipid.

What is the purpose for including the icl1/icl2 mutant? This experiment is also not included in the data quantification.

5) Discussion of Previously Published Data: Similar alterations on the *M. tuberculosis* cell wall and release of cell wall components to the milieu when exposed to physiological concentrations of human lung surfactant have been already described. The same is applicable to the slower replication rate in ATs (an intracellular killing in macrophages) after *M. tuberculosis* exposure to human lung surfactant. Although two different systems, authors need to contrast their findings with these reported ones in their discussion. In addition, it is not clear how many times this was performed. Statistics are mentioned on the figure legends, but there are not stats in the figure.

[Editors' note: further revisions were suggested prior to acceptance, as described below.]

Thank you for resubmitting your work entitled "A lung-on-chip model reveals an essential role for alveolar epithelial cells in controlling bacterial growth in early TB" for further consideration by *eLife*. Your revised article has been evaluated by Wendy Garrett (Senior Editor) and a Reviewing Editor.

The manuscript has been improved but there are some remaining issues that need to be addressed with text edits before acceptance, as outlined below:

1) Similar to the changes made in the Results section, the authors should change the title from “early tuberculosis” to “early *Mycobacterium tuberculosis* infection”.

2) The authors repeatedly refer to bacteria as intracellular, including statements like "which provides strong indirect evidence that Mtb growth on-chip must be intracellular". While a few confocal images are provided to highlight some intracellular bacteria, it is mostly an assumption that the bacteria are intracellular, and it would be more appropriate to be conservative in the interpretation and wording used in the manuscript.

3) The attenuation of the Esx-1 mutant is overstated. Specifically, the authors write "Macrophages are less permissive than ATs for intracellular growth of ESX-1 deficient Mtb under both NS and DS conditions". However, Figure 4A, which compares WT and Esx-1 deficient Mtb in NS macrophages demonstrates no difference in growth rates. Thus, the Results and Discussion should be revised to reflect this observation.

4) In Figure 1N, Q: Labeling shows AEC, but should it be AT?

5) In Figure 2—figure supplement 1: Please explain colors in the figure legends.

---

## [Author Response]

Essential revisions:1) Controls for the Model: What quality control is done for each experiment to determine the ratio of type I and type II AECs in each chip set up for each experiment? This is of particular importance because the authors do not show any images where they stain for both type I and type II AECs in the same chip. Do the authors have images stained for both type of cells to illustrate the composition of each chip? After Figure 1, what staining is done to confirm the DS cells decrease proSPC expression for each experiment? As this is a new model, better images showing well-defined cells and quantification of the number of alveolar epithelial cells type I and II, number of macrophages, numbers of endothelial cells, bacteria per cell, etc. would be helpful to evaluate the model. This applies particularly to Figures 1, 2 and Figure 3 supplement.

We acknowledge the concern of the reviewers regarding a careful characterization of the cell types to ensure reproducible results. We cannot characterize ATII and ATI for each live-imaging experiment for Mtb infections because the throughput for such a complex model is low and setting up multiple LoCs to a consistent standard requires extensive work. However, we have expanded the characterization of the DS and NS populations in vitro in liquid culture as well as in situ on-chip to be confident that this phenotype is reproducible.

Characterization of NS and DS ATs: In the revised manuscript, we expanded the list of Type II AT markers in the qRT-PCR characterization to include all surfactant proteins (*Sftpa*, *Sftpb*, *Sftpc*, *Sftpd*). We also included additional type I markers such as Podoplanin (*Pdpn*), and other markers such as Calveolin-1 (*Cav1*) and Insulin Growth Factor Binding Protein 2 (*Igfbp2*) that have been reported in the literature to be type I AT markers. These data are shown in a new Figure 1—figure supplement 1. Interestingly, we observed that the DS population had reduced expression of type I AT markers that could be linked to function at the air-liquid interface (such as *Aqp5* and *Pdpn*) in comparison to NS cells, but had increased expression of other type I AT markers (*Cav1* and *Ifgbp2*), consistent with an increase in terminally differentiated type I AT cells in the DS population.

In situ characterization of NS and DS LoCs: In the revised manuscript, we have also expanded the in situ characterization of the epithelial layer of the LoCs. Since both Pdpn and Aqp5 are membrane proteins, we were unable to visualize them on the same LoC that had been permeabilized for immunostaining for pro-SPC. We therefore imaged multiple, large fields of view via confocal microscopy to quantify the size and volume of lamellar bodies. Z projections are shown in a new panels H and I in Figure 1 and in Figure 1—figure supplement 2. We quantified the number and volume of pro-SPC+ lamellar bodies in an LoC reconstituted with NS and DS ATs, and these new data are included in new panels J and K in Figure 1. We also show that these differences in pro-SPC intensity are retained for 6 days at the air-liquid interface in Figure 1—figure supplement 2. Lastly, we also extracted the RNA from the apical channel of one NS and DS LoC and characterized the epithelial cell population via qRT-PCR. (new Figure 1L). This analysis confirmed that the DS LoCs have decreased expression of *Abca3* and some surfactant protein genes. We did not observe differences in expression of type I AT markers or *cxcl15*, a chemokine expressed exclusively by lung epithelial cells. Our results suggest that the primary difference between the NS and DS LoCs is primarily restricted to changes in surfactant production and levels.

Characterization of cell numbers: In the revised manuscript, we have provided a quantification of cell densities of ATs, endothelial cells, and macrophages on a typical LoC. These new data are included in a new Table 1.

The description of Figure 1 in the text has been changed in numerous places to account for the additional data in Figure 1—figure supplements 1 and 2, as well as Figure 1I-L, as discussed above.

2) Cell Identification: The authors focus on the difference in surfactant gene expression in the newly isolated AECs (NS) versus in vitro passaged AECs (DS), but they also observe that aqp5 is downregulated. In fact, the data support that the cells are de-differentiating during passage in culture, which will have multiple effects on the cells, not just surfactant production. This needs to be further studied in detail to ensure that this cell is not an artifact produced by multi-passaging in vitro. After loss of those markers, how do the authors confirm they still have type I and type II AECs in their cultures? Is there microscopy data with other markers that are retained in the AECs? The authors should use several AT-IIs and AT-Is markers to be certain that the DS cell monolayers indeed still are AECs. Surfactant protein C, although used as a marker for AT-IIs, is a soluble protein that has been shown to interact with many cells within a cellular system. A correlation between SPTPC and AQP5 expression over time is also necessary as points out the differentiation of AT-IIs to AT-Is, a key feature of the role of AT-IIs as progenitors of AT-Is.

Our response to Essential Revisions point 1 above already covers a substantial amount of the ground for this point. In the revised manuscript, we have characterized both freshly isolated ATs obtained directly from Cell Biologics and the in vitro passaged ATs with an expanded panel of type II and type I AT markers via qRT-PCR. The new data for this are included in new Figure 1—figure supplement 1A. This shows that passage lowers expression of type II markers linked to surfactant production and secretion as well as type I AT membrane proteins, but increases expression of genes such as *Cav1* and *Igfbp2* that are reported to be expressed in terminally differentiated type I ATs. We therefore believe that these results are consistent with the in vitro type II to type I differentiation and subsequent proliferation of type I ATs. In contrast, on-chip at the air-liquid interface, expression of type I AT markers was not different between the NS and DS LoCs, whereas the expression of type II AT markers such as *Abca3*, *Sftpa*, and *Sftpc* remained lower (Figure 1L). This correlated well with immunostaining for pro-SPC+ lamellar bodies (Figure 1J-K). We also show that the differences in pro-SPC+ lamellar body size and intensity are maintained for up to 6 days at the air-liquid interface (Figure 1—figure supplement 2).

3) Analysis of Mtb Infection and Intracellular versus Extracellular Infection: The authors never stain for type I or type II AECs after infection and make the conclusion that the bacteria are within type II cells based on the absence of macrophage staining. However, the bacteria may not even be inside a cell, or the AECs could be dying during infection. On a related note, there are no data presented that show that type I cells are not infected in the lung on chip system with Mtb. Thus, higher resolution images, with clear colocalization and z-stacks, would increase the confidence in the results. The images and analysis provided do not distinguish between extracellular and intracellular growth. Mtb can form micro-colonies on the cell surface of alveolar epithelial cells and thus, the intracellular growth that they are reporting could be extracellular growth. The authors could treat the system with an antibiotic to kill extracellular Mtb attached to the alveolar epithelial cell surface. Also, can the authors discern between Mtb that are not growing vs. Mtb that are dead?

We believe the reviewers might have misconstrued our claims from the paper. On-chip we only claim that ATs can be infected; we do not distinguish between infection of Type I or Type II cells because we do not have markers for cell-type characterization during live-cell imaging.

Intracellular vs extracellular growth and identifying macrophage infections: The use of fluorescently labelled macrophages allowed us to unambiguously identify bacteria that co-localise with macrophages over many hours on-chip. We think it is a fair assumption that these bacteria are internalized by the macrophages as one would expect from phagocytic cells, and in many cases we can see the shape of the bacterial structures change due to corresponding changes in macrophage cell shape. This assumption has also been born out in numerous examples of macrophage infection from confocal imaging of fixed samples. We highlight one example in the new Figure 2—figure supplement 1 and additional examples are also shown in Figure 3—figure supplement 1D-G.

Verification of infection of AT cells: First, images from confocal microscopy characterization of both uninfected and infected LoCs shows clearly the presence of a confluent layer of epithelial cells on the LoC (Figure 1E-I and Figure 1—figure supplement 2, Figure 2—figure supplement 1). In the case of the infected LoCs, this confluent layer is retained for at least the first four days post infection (Figure 2—figure supplement 3) (and in many cases even later) which corresponds to the period of analysis of growth on chip. Further, the confocal images also demonstrate two clear examples of a intracellular growth in ATs. The first shows small number of bacteria, and the second shows a larger cord that forms intracellularly (Figure 2—figure supplement 1, Video 4, 5). This substantially strengthens the argument for our assumption that Mtb that do not co-localise with macrophages are growing intracellularly in ATs.

Measures taken to avoid analysis of growth of bacteria on cell debris: It is indeed likely that Mtb might grow on cell debris at later timepoints subsequent to host cell death and the formation of Mtb clumps. We therefore begin infection with a single cell suspension, and we have been careful to analyse growth rates only in first contact host cells between 2-4 days post-infection. We have clarified this point in the Results.

In summary, although we cannot unambiguously rule out the occurrence of extracellular Mtb growth between 2-4 dpi for each datapoint in Figures 2-4, a scan of multiple infected cells via confocal microscopy at 4 dpi revealed only isolated examples where there could be some doubt as to the intracellular nature of growth, and certainly not at a level that would bias the results. We additionally show in Figure 2—figure supplement 3 that axenic Mtb growth in the ALI media (the presumed source of nutrients for extracellular bacteria) is very slow, and certainly could not support the rapid growth we observe in the absence of surfactant. We have included sentences in the text to mention this point. Data for the Δ*icl1*Δ*icl2* mutant (discussed in detail under Essential Revisions point 5) further corroborates this.

Administration of antibiotics: We do not see how we could administer antibiotics to bacteria growing extracellularly on top of the epithelial cells without disrupting the air-liquid interface, a key feature of the model.

Classification of non-growing vs dead Mtb: We are not able to clearly distinguish Mtb that are dead in the absence of a reliable live/dead marker; hence, we have restricted ourselves to calling this population “non-growing”. We have added a sentence in the text to make this clear.

4) Studies with the Mtb Mutants: The data arguing for attenuation of Esx-1 mutant Mtb in AECs and macrophages are not strong, and the authors do not actually make a direct statistical comparison between appropriate groups (i.e. AEC NS WT vs Esx-1, or Mac NS WT vs Esx-1). For example, it appears that the mean/median growth rate of WT Mtb in macrophages is ~0.25hr-1, which appears roughly the same for Esx-1 mutant Mtb in the same cells. There may be a difference under DS conditions, but since the comparisons are not made directly it is impossible to know. The authors state that their data with the Esx1 mutant "demonstrates that ESX-1 secretion is necessary for rapid intracellular growth in the absence of surfactant, consistent with the hypothesis that surfactant may attenuate Mtb growth by depleting ESX-1 components on the bacterial cell surface". This seems like quite a jump in interpretation of the data since the Esx1 mutant is likely attenuated for many reasons, and this attenuation is dominant to any effect that surfactant is having. The authors also show that PDIM levels are not different in the presence or absence of surfactant, and this is an Esx1 dependent lipid.

The data for the direct statistical comparison between the WT and ESX-1 deficient mutant are shown in Figure 4A-D and demonstrates that the ESX-1 deficient mutant does not grow as rapidly as the WT in the absence of surfactant. We therefore believe that our assertion in the text regarding the need for ESX-1 secretion for rapid intracellular growth in the absence of surfactant is supported by the data. Although ESX-1 secretion and PDIM secretion share transcriptional regulators such as EspR (Blasco et al., PMID: 22479184), ESX-1 secretion has been shown to be independent of PDIM secretion (Quigley et al. PMID: 28270579) and PDIM secretion is possible in ESX-1 deficient strains (Augenstreich et al., PMID: 28095608). We have also verified that the *ESX-1* strain does produce and secrete PDIM (Figure 4—figure supplement 1) and have added text to clarify this. Our data in Figure 5–figure supplement 2 shows that Curosurf treatment does not appear to remove PDIM from the cell surface from WT Mtb. In our view, this reinforces the point that a key reason for the reduced growth potential of the ESX-1 deficient mutant relative to WT Mtb in DS LoCs but not the NS LoCs is due the depletion of ESX-1 components by surfactant.

What is the purpose for including the icl1/icl2 mutant? This experiment is also not included in the data quantification.

The Δ*icl1*Δ*icl2* mutant serves two purposes. First it serves as a validation of the model; the Δ*icl1*Δ*icl2* mutant has a severely attenuated phenotype in vivo but is not attenuated for axenic growth in vitro so this is a clear benchmark that our model must meet. We have provided additional images taken with a confocal microscopy at 6 days post infection (Figure 3—figure supplement 1D-G), which show that the bacteria are intracellular, and remain as single bacteria and have not grown into clumps or cords. Second, this mutant has been shown not to grow intracellularly, but we show that it grows just as rapidly as WT Mtb in the ALI media that is perfused through the vascular channel of the chips (Figure 2—figure supplement 3), and which presumably would be the source of nutrients to these bacteria were they to grow extracellularly. The lack of growth provides another validation, albeit indirectly, that the bacterial growth in the LoC model occurs intracellularly. We have included sentences in the text to make this point.

We do not observe on-chip growth for the Δ*icl1*Δ*icl2* strain and show that these bacteria remain as single cells for a period of up to 6 days. The low fluorescence intensities of single Mtb bacteria during live-cell imaging made it difficult to obtain statistics to calculate the growth rate for this mutant. We therefore did not include it in the data quantification.

5) Discussion of Previously Published Data: Similar alterations on the M. tuberculosis cell wall and release of cell wall components to the milieu when exposed to physiological concentrations of human lung surfactant have been already described. The same is applicable to the slower replication rate in ATs (an intracellular killing in macrophages) after M. tuberculosis exposure to human lung surfactant. Although two different systems, authors need to contrast their findings with these reported ones in their discussion. In addition, it is not clear how many times this was performed. Statistics are mentioned on the figure legends, but there are not stats in the figure.

We thank the reviewers for comprehensively summarizing the various interactions of surfactant with Mtb that have already been reported. While we believe we cited these studies already in the previous draft, it is true that we did not discuss these observations in sufficient detail. We have therefore added sentences in the text to address this.

[Editors' note: further revisions were suggested prior to acceptance, as described below.]

The manuscript has been improved but there are some remaining issues that need to be addressed with text edits before acceptance, as outlined below:1) Similar to the changes made in the Results section, the authors should change the title from “early tuberculosis” to “early Mycobacterium tuberculosis infection”.

If possible, we would like to retain the title as it currently stands (“early tuberculosis”) so that citations for the preprint on biorxiv are appropriately attributed to this manuscript. In addition, even the use of “early tuberculosis” put the title over the 150 character limit of the *eLife* format, and the title suggested by the reviewers will go considerably over this limit. However, we have incorporated the change in the revised manuscript, although this will not be reflected in the title section of the online submission as the system does not allow a title longer than the character limit.

2) The authors repeatedly refer to bacteria as intracellular, including statements like "which provides strong indirect evidence that Mtb growth on-chip must be intracellular". While a few confocal images are provided to highlight some intracellular bacteria, it is mostly an assumption that the bacteria are intracellular, and it would be more appropriate to be conservative in the interpretation and wording used in the manuscript.

We have toned down the claims in the manuscript.

3) The attenuation of the Esx-1 mutant is overstated. Specifically, the authors write "Macrophages are less permissive than ATs for intracellular growth of ESX-1 deficient Mtb under both NS and DS conditions". However, Figure 4A, which compares WT and Esx-1 deficient Mtb in NS macrophages demonstrates no difference in growth rates. Thus, the Results and Discussion should be revised to reflect this observation.

The reviewers have raised a complex point, and we respectfully disagree with their interpretation. There are three variables in this system for which comparisons are made: bacterial strain, host cell type, and the surfactant conditions. The comparison across surfactant conditions (keeping host cell type and bacterial strain constant) is shown in Figure 3. The reviewers have referred to the first half of a sentence that compares growth of the WT or the ESX-1 deficient strain in macrophages relative to ATs under different surfactant conditions (a *comparison across host cell types* keeping bacterial strain and surfactant conditions constant) but cited data from Figure 4A. The data that is relevant to the sentence referred to by the reviewers is in Figure 3—figure supplement 2, where for both strains, growth is slower in macrophages vs. ATs (sub-panel A for WT and sub-panel C for ESX-1 deficient Mtb). However, ESX-1 deficient bacteria show these cell-type specific differences even in DS conditions (sub-panel D) whereas WT do not (sub-panel B). It is these differences that we sought to capture in the sentence quoted by the reviewers. Finally, the comparison across bacterial strains (keeping surfactant conditions and host cell constant) are shown in Figure 4A-D. We agree with the reviewers that Figure 4A does show that there is no additional attenuation of the ESX-1 relative to the WT in NS macrophages. Rather, ESX-1 deficient bacteria are unable to capitalize on the surfactant deficiency and grow as fast as their WT counterparts as we show in Figure 4B and 4D.

As it appears that the current phrasing of the sentence highlighted by the reviewer and the subsequent sentences cause confusion, we have rewritten them to elaborate on the points stated above. We hope the revised text which follows the order of comparison across surfactant conditions, followed by bacterial strains, and then by host cell-type is clearer.

Results

“ In comparison to wild-type Mtb, whose intracellular growth rate is strongly dependent on surfactant levels (Figure 3D-G), intracellular growth of the 5’Tn::*pe35* strain ^22^ that is deficient in ESX-1 secretion but retains PDIM secretion (Figure 4—figure supplement 1) is largely independent of surfactant levels (Figure 3H-K). Under NS conditions, a greater fraction of ESX-1 deficient bacteria are “non-growing” in ATs (Figure 3I). Under DS conditions, the ESX-1 deficient strain is unable to grow as rapidly as wild-type in both macrophages and ATs (Figure 4B, D) and a fraction (ca. 12%) of ESX-1 deficient bacteria are non-growing in macrophages (Figure 3K). This attenuation in DS conditions is evident by visual inspection at 6 days post infection (Figure 3B vs 3A). Macrophages are less permissive than ATs for intracellular growth of ESX-1 deficient Mtb under both NS and DS conditions (Figure 3—figure supplement 2 C, D). In contrast, wild-type Mtb, grows more slowly in macrophages than in ATs only under NS but not DS conditions (Figure 3—figure supplement 2 A,B). Overall, attenuation of ESX-1 deficient bacteria relative to WT is not rescued by surfactant deficiency.”

Discussion

We have toned down the text by removing the adjective “good” in the sentence to describe the agreement between the LoC model and in vivo experiments in relation to the ESX-1 deficient strain.

4) In Figure 1N, Q: Labeling shows AEC, but should it be AT?

We apologise for the inadvertent error, this has been corrected.

5) In Figure 2-Figure 1 Supplement: Please explain colors in the figure legends.

The information for the colors has been incorporated in the figure legend for Figure 2—figure supplement 1, as well as in Figure 1—figure supplement 2.